# High Probability Bounds for Non-Convex Stochastic Optimization with Momentum

**Shaojie Li[1], Pengwei Tang[2], Bowei Zhu[2], Yong Liu[2],\***
[1]College of Design and Engineering, National University of Singapore, Singapore
[2]Gaoling School of Artificial Intelligence, Renmin University of China, Beijing, China
`li_sj@nus.edu.sg`, `{tangpwei,bowei.zhu,liuyonggsai}@ruc.edu.cn`

## Abstract

Stochastic gradient descent with momentum (SGDM) is widely used in machine learning, yet high-probability learning bounds for SGDM in non-convex settings remain scarce. In this paper, we provide high-probability convergence bounds and generalization bounds for SGDM. First, we establish such bounds for the gradient norm in the general non-convex case. The resulting convergence bounds are tighter than existing theoretical results, and the obtained generalization bounds seem to be the first for SGDM. Next, under the Polyak-Łojasiewicz condition, we derive bounds for the function-value error instead of the gradient norm, and the corresponding learning rates are faster than in the general non-convex case. Finally, by additionally assuming a mild Bernstein condition on the gradient, we obtain even sharper generalization bounds whose learning rates can reach $\widetilde{\mathcal{O}}(1/n^2)$ in the low-noise regime, where $n$ is the sample size. Overall, we provide a systematic study of high-probability learning bounds for non-convex SGDM.

## 1 Introduction

Stochastic optimization plays an essential role in modern statistics and machine learning, as many learning problems can be cast as stochastic optimization tasks. Over the past decades, there has been substantial progress in the development of stochastic optimization algorithms, among which stochastic gradient descent with momentum (SGDM) has attracted particular attention due to its simplicity and low per-iteration computational cost (Goodfellow et al., 2016; Li & Orabona, 2020). As a fundamental algorithm for stochastic optimization, SGDM has been remarkably successful in natural language understanding, computer vision, and speech recognition (Krizhevsky et al., 2012; Hinton et al., 2012; Sutskever et al., 2013).

Typically, SGDM augments stochastic gradient descent (SGD) with a momentum term in the update rule, i.e., it uses the difference between the current and previous iterates. The intuition is that, when the recent update direction is informative about the local descent direction, SGDM can leverage this inertial signal—scaled by the momentum parameter—to smooth stochastic gradients rather than relying solely on the instantaneous gradient at the current iterate, as in plain SGD. Much of the state-of-the-art empirical performance in deep learning has been achieved using SGDM (Huang et al., 2017; Howard et al., 2017; He et al., 2016; Kim et al., 2021a). Yet, from a theoretical standpoint, the analysis of learning bounds for SGDM remains relatively underdeveloped (Li et al., 2022; Li & Orabona, 2020).

The learning performance of SGDM can be studied from two complementary perspectives: *convergence bounds* and *generalization bounds*. Convergence bounds focus on how well the algorithm optimizes the empirical risk, whereas generalization bounds quantify how the learned model performs on unseen test data. From the convergence perspective, existing analyses of SGDM or deterministic gradient descent with momentum (DGDM) in non-convex settings are mostly in *expectation* (Ochs et al., 2014; 2015; Ghadimi et al., 2015; Lessard et al., 2016; Yang et al., 2016; Wilson et al., 2021; Gadat et al., 2018; Orvieto et al., 2020; Can et al., 2019; Li et al., 2022; Yan et al., 2018; Liu et al., 2020), to mention only a few. However, expected bounds do not rule out

---

*Corresponding Author.

the possibility of extremely bad outcomes (Li & Orabona, 2020; Liu et al., 2023). Moreover, in practical large-scale applications, the training procedure is typically run only once, since it can be very time-consuming. For such single-run performance, high-probability bounds are more informative than expectation bounds (Harvey et al., 2019). To the best of our knowledge, there are two works that provide high-probability convergence bounds for SGDM (Li & Orabona, 2020; Cutkosky & Mehta, 2021). Specifically, Cutkosky & Mehta (2021) assume that the gradient noise satisfies a $\theta$-order moment condition with $\theta \in (1, 2]$ and obtain a convergence rate of order $\widetilde{\mathcal{O}}\big(T^{-\frac{\theta-1}{3\theta-2}}\big)$ for the gradient norm, where $T$ denotes the number of iterations. Li & Orabona (2020) establish a convergence bound of order $\widetilde{\mathcal{O}}(1/\sqrt{T})$ for the squared gradient norm under sub-Gaussian gradient noise. As discussed in Li et al. (2022), it is unclear whether this rate can be improved or extended to more general noise models beyond the sub-Gaussian case. Overall, the convergence rates in Li & Orabona (2020); Cutkosky & Mehta (2021) are relatively slow, and, importantly, *no* generalization bounds are provided in either work.

From the generalization perspective, existing results for SGDM and DGDM are even scarcer. Ong (2017); Chen et al. (2018) derive expected generalization error bounds for DGDM with a specific quadratic loss by using algorithmic stability (Bousquet & Elisseeff, 2002; Hardt et al., 2016). Their analysis, however, does not extend easily to general loss functions. It is conjectured in Chen et al. (2018) that their uniform stability bound might also hold for general convex losses. Motivated by this conjecture, Ramezani-Kebrya et al. (2024) study generalization error bounds for SGDM with general loss functions. Somewhat surprisingly, they construct a counterexample showing that, even for convex loss functions, the uniform stability gap (in expectation, over the internal randomness of the algorithm) of SGDM run for multiple epochs can diverge. In a related direction, Attia & Koren (2021) show that, in the general convex case, the uniform stability gap of deterministic Nesterov's accelerated gradient (NAG) can diverge exponentially fast with the number of iterations. We emphasize that uniform stability is only a *sufficient* condition for generalization; it remains unclear how weaker stability notions (such as on-average stability (Shalev-Shwartz et al., 2010)) behave for SGDM. Overall, there are significant obstacles to establishing general generalization guarantees for SGDM, especially for broad classes of loss functions. Furthermore, as in the convergence analysis of SGDM, high-probability generalization bounds are substantially more challenging to derive than expectation-based bounds (Bousquet et al., 2020; Bassily et al., 2020; Feldman & Vondrak, 2019).

Therefore, both high-probability convergence bounds and high-probability generalization bounds for SGDM remain far from fully understood. Motivated by the above limitations, this paper aims to establish such bounds for SGDM, with a particular focus on non-convex settings. For brevity, we will refer to all bounds on the performance of the learned model on test data (including generalization error bounds and excess-risk bounds) simply as *generalization bounds*. Our main contributions can be summarized as follows.

- At a high level, we study the case where the stochastic gradient noise follows a sub-Weibull distribution (Vladimirova et al., 2019; 2020; Kuchibhotla & Chakrabortty, 2018), which generalizes the sub-Gaussian noise considered in Li & Orabona (2020) to potentially heavier-tailed distributions. Our learning bounds under this assumption reveal how the rates of convergence and generalization change as one moves from sub-Gaussian / sub-exponential (light-tailed) noise to heavy-tailed noise with exponential-type tails.

- We first provide a high-probability analysis of SGDM in the general non-convex case. In this setting, we derive convergence bounds of order $\widetilde{\mathcal{O}}(1/T^{1/2})$ and generalization bounds of order $\widetilde{\mathcal{O}}\big(d^{1/2}/n^{1/2}\big)$ for the squared gradient norm, where $d$ is the dimension and $n$ is the sample size. The convergence bounds are tighter than those in related work, and the generalization bounds seem to be the *first* such results for SGDM.

- We next analyze SGDM under the Polyak–Łojasiewicz condition for non-convex objectives. In this case, we obtain sharper convergence bounds of order $\widetilde{\mathcal{O}}(1/T)$. Furthermore, these bounds are established for the *last iterate* of SGDM and for the *function-value error*, rather than for the average iterate and gradient norm considered in the general non-convex case. In addition, we derive generalization bounds of faster order $\widetilde{\mathcal{O}}\big(\frac{d+\log(1/\delta)}{n}\big)$ for SGDM, which, to our knowledge, have not been previously available.

- Finally, we impose a mild Bernstein condition on the gradient. Under this additional assumption, we improve the generalization bound of order $\widetilde{\mathcal{O}}\big(\frac{d+\log(1/\delta)}{n}\big)$ to a bound of order

$\widetilde{\mathcal{O}}(1/n^2 + F^*/n)$, where $F^*$ denotes the optimal population risk. In the low-noise regime where $F^*$ is small, this bound yields a faster learning rate of order $\widetilde{\mathcal{O}}(1/n^2)$, showing a tighter dependence on the sample size $n$. Another attractive feature of this bound is that the dimension $d$ no longer appears, allowing it to easily incorporate massive neural networks that are often high-dimensional.

In summary, by considering increasingly strong structural conditions on the objective function (from general non-convexity, to PL, to PL plus a Bernstein condition), we establish a hierarchy of improved learning bounds with different rates. This provides a systematic picture of the high-probability learning guarantees for SGDM from both convergence and generalization perspectives.

The rest of the paper is organized as follows. Preliminaries are presented in Section 2. Our main results are stated in Section 3. We conclude in Section 4. Numerical experiments are reported in Section A. Appendix B, together with Table 1, summarizes our main results and the most relevant related bounds of SGDM. All proofs are deferred to the Appendix.

## 2 PRELIMINARIES

### 2.1 NOTATIONS

Let $\mathcal{X} \subseteq \mathbb{R}^d$ be the parameter space and let $\mathbb{P}$ be a probability measure on a sample space $\mathcal{Z}$. Let $f : \mathcal{X} \times \mathcal{Z} \to \mathbb{R}_+$ be a (possibly non-convex) loss function. We consider the stochastic optimization problem

$$\min_{\mathbf{x} \in \mathcal{X}} F(\mathbf{x}) := \mathbb{E}_{z \sim \mathbb{P}}[f(\mathbf{x}; z)],$$

where $F$ is referred to as the *population risk* and $\mathbb{E}_{z \sim \mathbb{P}}$ denotes expectation with respect to (w.r.t.) the random variable $z$ drawn from $\mathbb{P}$.

In practice, the distribution $\mathbb{P}$ is unknown and we only observe a dataset $S = \{z_1, \ldots, z_n\}$ drawn independently and identically (i.i.d.) from $\mathbb{P}$. One typically optimizes the *empirical risk*

$$\min_{\mathbf{x} \in \mathcal{X}} F_S(\mathbf{x}) := \frac{1}{n} \sum_{i=1}^n f(\mathbf{x}; z_i).$$

To optimize $F_S$, SGDM has been widely adopted (Polyak, 1964; Qian, 1999; Sutskever et al., 2013; Li & Orabona, 2020). In this work we focus on Polyak's momentum, also known as the heavy-ball algorithm or *classical* momentum, which is arguably the most popular form of momentum in current machine learning practice (Liu et al., 2020). The pseudocode of SGDM (Polyak's momentum) is given in Algorithm 1. The vanilla SGD update is

$$\mathbf{x}_{t+1} = \mathbf{x}_t - \eta_t \nabla f(\mathbf{x}_t; z_{j_t}).$$

In Step 3 of Algorithm 1, SGDM introduces a momentum vector $\mathbf{m}_{t-1}$ and forms a momentum term weighted by a parameter $\gamma$ to adjust the gradient estimate $\nabla f(\mathbf{x}_t; z_{j_t})$ of SGD. In Step 4, SGDM then updates the iterate via

$$\mathbf{x}_{t+1} = \mathbf{x}_t - \mathbf{m}_t.$$

Equivalently, the SGDM update can be written as

$$\mathbf{x}_{t+1} = \mathbf{x}_t - \eta_t \nabla f(\mathbf{x}_t; z_{j_t}) + \gamma(\mathbf{x}_t - \mathbf{x}_{t-1}).$$

We now introduce some notation. Let $B = \sup_{z \in \mathcal{Z}} \|\nabla f(\mathbf{0}; z)\|$, where $\nabla f(\cdot; z)$ denotes the gradient of $f$ w.r.t. the first argument and $\|\cdot\|$ denotes the Euclidean norm. For any $R > 0$, we define $B(\mathbf{x}_0, R) := \{\mathbf{x} \in \mathbb{R}^d : \|\mathbf{x} - \mathbf{x}_0\| \leq R\}$ which denotes a ball with center $\mathbf{x}_0 \in \mathbb{R}^d$ and radius $R$. Let $\mathbf{x}(S) \in \arg\min_{\mathbf{x} \in \mathcal{X}} F_S(\mathbf{x})$ and $\mathbf{x}^* \in \arg\min_{\mathcal{X}} F(\mathbf{x})$. We write $a \asymp b$ if there exist universal constants $c, c' > 0$ such that $ca \leq b \leq c'a$. Throughout the paper we use standard order-notation such as $\mathcal{O}(\cdot)$ and $\widetilde{\mathcal{O}}(\cdot)$.

### 2.2 ASSUMPTIONS

In this subsection we collect the assumptions that will be invoked in our main theorems.

**Assumption 2.1.** The differentiable function $f$ is (possibly) non-convex and, for any $z \in \mathcal{Z}$, the mapping $\mathbf{x} \mapsto f(\mathbf{x}; z)$ is $L$-smooth, i.e., for every $\mathbf{x}_1, \mathbf{x}_2$:

$$\|\nabla f(\mathbf{x}_1; z) - \nabla f(\mathbf{x}_2; z)\| \leq L \|\mathbf{x}_1 - \mathbf{x}_2\|,$$

where $\nabla$ is the gradient operator and $\|\cdot\|$ is the Euclidean norm.

*Remark* 2.2. Further properties of smooth functions are collected in Lemma C.7.

**Assumption 2.3.** The gradient at $\mathbf{x}^*$ satisfies a Bernstein-type moment condition: there exists $B_* > 0$ such that for all integers $k$ with $k \geq 2$,

$$\mathbb{E}_z \left[ \|\nabla f(\mathbf{x}^*; z)\|^k \right] \leq \frac{1}{2} k! \, \mathbb{E}_z \left[ \|\nabla f(\mathbf{x}^*; z)\|^2 \right] B_*^{k-2}.$$

*Remark* 2.4. The Bernstein condition is standard in learning theory. As shown in Wainwright (2019), for a random variable $X$ with mean $\mu = \mathbb{E}[X]$ and variance $\sigma^2 = \mathbb{E}[X^2] - \mu^2$, we say that $X$ satisfies the Bernstein condition with parameter $b$ if for all integers $k \geq 2$,

$$\mathbb{E}\left[ (X - \mu)^k \right] \leq \frac{1}{2} k! \, \sigma^2 b^{k-2}.$$

The Bernstein condition is essentially equivalent to $X$ being sub-exponential; see the discussion in Remark 4 of Lei (2020). Classical sub-Gaussian and sub-exponential distributions satisfy this condition, since their $k$-th moments are controlled by the second moment. In this sense, the Bernstein condition is quite mild and, for instance, weaker than assuming that $X$ is almost surely bounded. Assumption 2.3 simply applies this Bernstein condition to the random variable $\|\nabla f(\mathbf{x}^*; z)\|$: it is weaker than assuming that $\|\nabla f(\mathbf{x}; z)\|, \forall \mathbf{x} \in \mathcal{X}$, is uniformly bounded, while the latter bounded-gradient assumption is widely used in stochastic optimization (Zhang et al., 2017).

**Assumption 2.5.** For all $S \in \mathcal{Z}^n$, and for some positive $G > 0$, the empirical risk satisfies

$$\eta_t \|\nabla F_S(\mathbf{x}_t)\| \leq G, \quad \forall t \in \mathbb{N}.$$

*Remark* 2.6. In the theoretical analysis of stochastic optimization, it is common to assume a uniformly bounded stochastic gradient,

$$\|\nabla f(\mathbf{x}; z)\| \leq G, \qquad \forall \mathbf{x} \in \mathcal{X}, \, \forall z \in \mathcal{Z},$$

which is sometimes referred to as the Lipschitz continuity of $f$ (Li et al., 2022; Li & Orabona, 2020). Assumption 2.5 is a relaxation of this bounded-gradient assumption: it multiplies the gradient norm of $F_S$ by the stepsize $\eta_t$ instead of bounding each stochastic gradient $\nabla f(\mathbf{x}_t; z)$. Since the stepsizes $\eta_t$ decrease to zero, the gradients of $F_S$ are allowed to grow. For typical decay rates $\eta_t = \mathcal{O}(t^{-1/2})$ or $\eta_t = \mathcal{O}(t^{-1})$ (Lei & Tang, 2021), Assumption 2.5 permits $\|\nabla F_S(\mathbf{x}_t)\|$ to grow at rates $\mathcal{O}(t^{1/2})$ and $\mathcal{O}(t)$, respectively.

In the next, we introduce the Polyak-Łojasiewicz (PL) condition.

**Assumption 2.7.** Fix a set $\mathcal{X}$ and let $f^* := \min_{\mathbf{x} \in \mathcal{X}} f(\mathbf{x})$. We say that a differentiable function $f : \mathcal{X} \to \mathbb{R}$ satisfies the Polyak–Łojasiewicz condition with parameter $\mu > 0$ on $\mathcal{X}$ if for all $\mathbf{x} \in \mathcal{X}$,

$$f(\mathbf{x}) - f^* \leq \frac{1}{2\mu} \|\nabla f(\mathbf{x})\|^2.$$

*Remark* 2.8. Fast rates cannot be achieved for free. The Polyak-Łojasiewicz condition is widely used in the optimization community to obtain fast convergence rates (Necoara et al., 2019; Karimi et al., 2016) and is one of the weakest curvature conditions to replace the strong convexity (Karimi et al., 2016). Many important models are known to satisfy a PL inequality, at least locally. Notable examples satisfying the PL condition include two-layer neural networks (Li & Yuan, 2017), matrix completion (Sun & Luo, 2016), dictionary learning (Arora et al., 2015), and phase retrieval (Chen & Candes, 2015). Kleinberg et al. (2018) provide empirical evidence that the (smoothed) loss of practical deep networks locally exhibits a one-point convexity property of PL type. More rigorously, Soltanolkotabi et al. (2018) analyze over-parameterized shallow networks with quadratic activations and prove that, in the interpolation regime where the training loss is zero, the empirical risk satisfies a PL inequality. These examples motivate our focus on studying SGDM under the PL curvature assumption.

---

**Algorithm 1** SGD with Momentum (SGDM)

---

**Require:** stepsizes $\{\eta_t\}_t$, dataset $S = \{z_1, ..., z_n\}$, and momentum parameter $0 < \gamma < 1$.
**Initializtion:** $\mathbf{x}_1 = \mathbf{0}$, $\mathbf{m}_0 = \mathbf{0}$,

1: **for** $t = 1, ..., T$ **do**
2:     sample $j_t$ from the uniform distribution over the set $\{j : j \in [n]\}$,
3:     update $\mathbf{m}_t = \gamma \mathbf{m}_{t-1} + \eta_t \nabla f(\mathbf{x}_t; z_{j_t})$,
4:     update $\mathbf{x}_{t+1} = \mathbf{x}_t - \mathbf{m}_t$.
5: **end for**

---

In our analysis we will apply Assumption 2.7 both to the empirical risk $F_S$ and to the population risk $F$. When studying optimization (training) performance, we assume that $F_S$ satisfies a PL inequality with parameter $\mu(S)$; when studying generalization and excess risk, we assume that $F$ satisfies a (possibly different) PL inequality with parameter $\mu$. We keep the notation $\mu(S)$ and $\mu$ separate to emphasize that the curvature at the sample level need not coincide exactly with that of the underlying population.

Finally, we specify an assumption on the noise of the stochastic gradient.

**Assumption 2.9.** The gradient noise $\nabla f(\mathbf{x}_t; z_{j_t}) - \nabla F_S(\mathbf{x}_t)$ satisfies

$$\mathbb{E}_{j_t}\left[ \exp(\|\nabla f(\mathbf{x}_t; z_{j_t}) - \nabla F_S(\mathbf{x}_t)\|/K)^{\frac{1}{\theta}} \right] \leq 2, \tag{1}$$

for some positive $K$ and $\theta \geq 1/2$.

*Remark* 2.10. Li & Orabona (2020) assume the sub-Gaussian-type condition

$$\mathbb{E}_{j_t}\left[ \exp\big(\|\nabla f(\mathbf{x}_t; z_{j_t}) - \nabla F_S(\mathbf{x}_t)\|^2/K^2\big) \right] \leq 2,$$

which ensures that the noise tails are dominated by those of a Gaussian distribution. In contrast, Assumption 2.9 generalizes this to a richer class of distributions, including sub-exponential noise (corresponding to $\theta = 1$) and even heavier-tailed noise ($\theta > 1$). Condition (1) is precisely the defining property of a *sub-Weibull* random variable (Vladimirova et al., 2020): a random variable $X$ satisfying $\mathbb{E}[\exp((|X|/K)^{1/\theta})] \leq 2$ for some $K > 0$ and $\theta \geq 1/2$ is called sub-Weibull with tail parameter $\theta$, and larger $\theta$ means heavier tails (Kuchibhotla & Chakrabortty, 2018). Hence, the learning bounds in this paper apply to a broad class of heavy-tailed gradient noise distributions. Our motivation for studying sub-Weibull gradient noise is twofold. First, it allows us to explicitly quantify how the convergence and generalization rates degrade when moving from sub-Gaussian/sub-exponential (light-tailed) noise to heavy-tailed noise with exponential-type tails. Second, a growing body of work provides empirical and theoretical evidence that the noise in stochastic optimization algorithms is often heavier-tailed than sub-Gaussian (Panigrahi et al., 2019; Madden et al., 2024; Gurbuzbalaban et al., 2021; Simsekli et al., 2019; Şimşekli et al., 2019; Zhang et al., 2020; 2019; Wang et al., 2021; Gurbuzbalaban & Hu, 2021).

## 3 MAIN RESULTS

This section presents our main theoretical results.

### 3.1 LEARNING BOUNDS IN THE GENERAL NON-CONVEX CASE

In the general non-convex case, we cannot guarantee that the algorithm finds a global minimizer, so we focus on approximate first-order stationary points. For the convergence analysis, we are interested in iterates $\mathbf{x}_t$ satisfying $\|\nabla F_S(\mathbf{x}_t)\|^2 \leq \epsilon$, while for generalization we consider $\|\nabla F(\mathbf{x}_t)\|^2 \leq \epsilon$. As is standard in the non-convex literature, we measure optimization and generalization performance via the average squared gradient norms $\frac{1}{T}\sum_{t=1}^{T}\|\nabla F_S(\mathbf{x}_t)\|^2$ and $\frac{1}{T}\sum_{t=1}^{T}\|\nabla F(\mathbf{x}_t)\|^2$, respectively.

#### 3.1.1 CONVERGENCE BOUNDS

We first provide high-probability convergence bounds for SGDM. These bounds characterize how the algorithm minimizes the empirical risk $F_S$.

**Theorem 3.1.** *Let $\mathbf{x}_t$ be the sequence of iterates generated by Algorithm 1. Set the stepsize as $\eta_t = ct^{-\frac{1}{2}}$, where $c \leq \frac{1}{4}\frac{(1-\gamma)^3}{3L - L\gamma}$.*

*(1). If $\theta = \frac{1}{2}$, suppose Assumptions 2.1 and 2.9 hold. Then for any $\delta \in (0,1)$, with probability $1 - \delta$,*

$$\frac{1}{T}\sum_{t=1}^{T}\|\nabla F_S(\mathbf{x}_t)\|^2 = \mathcal{O}\left(\frac{\log(1/\delta)\log T}{\sqrt{T}}\right).$$

*(2). If $\frac{1}{2} < \theta \leq 1$, suppose Assumptions 2.1, 2.5, and 2.9 hold. Then for any $\delta \in (0,1)$, with probability $1 - \delta$,*

$$\frac{1}{T}\sum_{t=1}^{T}\|\nabla F_S(\mathbf{x}_t)\|^2 = \mathcal{O}\left(\frac{\log^{2\theta}(1/\delta)\log T}{\sqrt{T}}\right).$$

*(3). If $\theta > 1$, suppose Assumptions 2.1, 2.5, and 2.9 hold. Then for any $\delta \in (0,1)$, with probability $1 - \delta$,*

$$\frac{1}{T}\sum_{t=1}^{T}\|\nabla F_S(\mathbf{x}_t)\|^2 = \mathcal{O}\left(\frac{\log^{\theta-1}(T/\delta)\log(1/\delta) + \log^{2\theta}(1/\delta)\log T}{\sqrt{T}}\right).$$

*Remark* 3.2. The bounds in Theorem 3.1 are all of order $\widetilde{\mathcal{O}}(1/\sqrt{T})$. The dependence on the tail parameter $\theta$ shows that larger $\theta$ leads to worse (slower) convergence, which matches the intuition that heavier-tailed gradient noise degrades optimization performance. We now compare these results with related work (Li & Orabona, 2020; Cutkosky & Mehta, 2021). Cutkosky & Mehta (2021) analyze a different algorithmic setting that combines gradient clipping, a variant of momentum (distinct from Polyak's momentum), and normalized gradient descent. Building on this line of work, Li & Liu (2023) further considers *clipping* variants of momentum. We primarily compare against (Cutkosky & Mehta, 2021), as its setting aligns more closely with this paper (*i.e., momentum without clipping*). The Theorem 2 of (Cutkosky & Mehta, 2021) establishes a convergence bound of order

$$\mathcal{O}\left(\frac{\log(T/\delta)}{T^{\frac{\theta-1}{3\theta-2}}}\right)$$

for $\frac{1}{T}\sum_{t=1}^{T}\|\nabla F_S(\mathbf{x}_t)\|$ under smoothness and a $\theta$-moment condition on the gradient, where $\theta \in (1, 2]$. In the case $\theta = 2$, this rate becomes $\widetilde{\mathcal{O}}(T^{-1/4})$. By Jensen's inequality,

$$\left(\frac{1}{T}\sum_{t=1}^{T}\|\nabla F_S(\mathbf{x}_t)\|\right)^2 \leq \frac{1}{T}\sum_{t=1}^{T}\|\nabla F_S(\mathbf{x}_t)\|^2,$$

so Theorem 3.1 implies the same $\widetilde{\mathcal{O}}(T^{-1/4})$ rate for $\frac{1}{T}\sum_{t=1}^{T}\|\nabla F_S(\mathbf{x}_t)\|$. Li & Orabona (2020) study Polyak's momentum and, in their Theorem 1, obtain a convergence bound of order

$$\mathcal{O}\left(\frac{\log(T/\delta)\log T}{\sqrt{T}}\right)$$

for $\frac{1}{T}\sum_{t=1}^{T}\|\nabla F_S(\mathbf{x}_t)\|^2$ under smoothness and sub-Gaussian gradient noise (i.e., $\theta = 1/2$). Since we also analyze Polyak's momentum, the comparison with Li & Orabona (2020) is more natural. Under the same assumptions, part (1) of Theorem 3.1 refines this to

$$\mathcal{O}\left(\frac{\log(1/\delta)\log T}{\sqrt{T}}\right).$$

Although this improvement is only logarithmic, it may be the strongest possible refinement in the general nonconvex setting we consider. For smooth nonconvex stochastic optimization with a first-order oracle and controlled noise, the rate $\mathcal{O}(1/\sqrt{T})$ in terms of the expected squared gradient norm is known to be optimal (up to logarithmic factors) (Arjevani et al., 2019). Consequently, under the same structural assumptions, any further progress can only affect constants and logarithms but not the leading $1/\sqrt{T}$ scaling. Theorem 2 in Li & Orabona (2020) further analyzes a variant of AdaGrad with Polyak's momentum, called delayed AdaGrad, whose stepsize does not depend on the current gradient (Li & Orabona, 2019). The corresponding convergence bound is of order

$$\max\left\{\mathcal{O}\left(\frac{d\,\log^{3/2}(T/\delta)}{\sqrt{T}}\right), \mathcal{O}\left(\frac{d^2\,\log^2(T/\delta)}{T}\right)\right\}.$$

When the dimension $d$ is small, this gives a rate of order $\mathcal{O}\big(d\,\log^{3/2}(T/\delta)/\sqrt{T}\big)$, which is clearly weaker than the dimension-free bounds in Theorem 3.1.

In the non-convex, stochastic setting, a clear separation between SGD and SGDM remains elusive (Li & Orabona, 2020; Zou et al., 2018), even at the empirical level. For example, Kidambi et al. (2018) provide theoretical and empirical evidence that standard momentum schemes—Polyak's momentum and Nesterov Accelerated Gradient—do not enjoy a universal acceleration guarantee in the stochastic regime. Even with optimally tuned hyperparameters, there exist instances where Polyak's momentum and Nesterov's method do not outperform vanilla SGD. In particular, when the batch size is small (e.g., 1), their performance is often nearly indistinguishable from, or even worse than, that of SGD. This batch-size-one regime is exactly the setting we study here, and our theory is consistent with these observations. A work (Li & Liu, 2022) derives high-probability convergence and generalization results for *SGD without momentum* under the same assumptions, yielding rates of the same order as those obtained here (up to constants and mild logarithmic factors). This paper *closes the theoretical gap for SGDM*: we show that the widely used momentum method, under the general nonconvex / PL / Bernstein assumptions, also enjoys high-probability convergence and generalization guarantees of comparable order to those known for SGD, so that SGDM has essentially the same theoretical performance as SGD under these conditions. A promising direction for future work is to extend our analysis to the large-batch regime and to investigate how the potential benefits of momentum depend on batch-induced noise reduction.

### 3.1.2 GENERALIZATION BOUNDS

We now present high-probability generalization bounds for SGDM, which quantify how well the learned models perform on the underlying data distribution.

**Theorem 3.3.** *Let $\mathbf{x}_t$ be the sequence of iterates generated by Algorithm 1. Set the stepsize as $\eta_t = ct^{-\frac{1}{2}}$, where $c \le \frac{1}{4}\frac{(1-\gamma)^3}{3L-L\gamma}$, and choose the number of iterations as $T \asymp n/d$.*

*(1). If $\theta = \frac{1}{2}$, suppose Assumptions 2.1 and 2.9 hold. Then for any $\delta \in (0,1)$, with probability $1-\delta$,*

$$\frac{1}{T}\sum_{t=1}^{T}\|\nabla F(\mathbf{x}_t)\|^2 \;=\; \mathcal{O}\bigg(\Big(\frac{d}{n}\Big)^{1/2}\log\Big(\frac{n}{d}\Big)\,\log^3\Big(\frac{1}{\delta}\Big)\bigg).$$

*(2). If $\frac{1}{2} < \theta \le 1$, suppose Assumptions 2.1, 2.5, and 2.9 hold. Then for any $\delta \in (0,1)$, with probability $1-\delta$,*

$$\frac{1}{T}\sum_{t=1}^{T}\|\nabla F(\mathbf{x}_t)\|^2 \;=\; \mathcal{O}\bigg(\Big(\frac{d}{n}\Big)^{1/2}\log\Big(\frac{n}{d}\Big)\,\log^{2\theta+2}\Big(\frac{1}{\delta}\Big)\bigg).$$

*(3). If $\theta > 1$, suppose Assumptions 2.1, 2.5, and 2.9 hold. Then for any $\delta \in (0,1)$, with probability $1-\delta$,*

$$\frac{1}{T}\sum_{t=1}^{T}\|\nabla F(\mathbf{x}_t)\|^2 \;=\; \mathcal{O}\bigg(\Big(\frac{d}{n}\Big)^{1/2}\Big(\log\Big(\frac{n}{d}\Big)\log^{2\theta+2}\Big(\frac{1}{\delta}\Big) \;+\; \log^{\theta-1}\Big(\frac{n}{d\delta}\Big)\log^2\Big(\frac{1}{\delta}\Big)\Big)\bigg).$$

*Remark* 3.4. The bounds in Theorem 3.3 are of order $\widetilde{\mathcal{O}}\big((d/n)^{1/2}\big)$, and again heavier tails (larger $\theta$) lead to slower rates. As in Theorem 3.1, when $\theta = 1/2$ Assumption 2.5 is no longer needed. As discussed in the introduction, algorithmic stability—in particular uniform stability—seems to fail for SGDM with general loss functions, since the uniform stability gap may diverge even in convex settings (Ramezani-Kebrya et al., 2024). This is consistent with the general principle that there is a trade-off between convergence speed and stability: faster-converging algorithms tend to be less stable, and vice versa (Chen et al., 2018). Our proof technique instead belongs to the *uniform convergence* approach (Xu & Zeevi, 2020; Lei & Tang, 2021), which shows that the empirical risks of all hypotheses in a class converge uniformly to their population risks. In the general non-convex case, a dependence on the ambient dimension $d$ is typically unavoidable for such uniform convergence bounds (Xu & Zeevi, 2020; Lei & Tang, 2021), which is reflected in the $d$-dependence in Theorem 3.3. We emphasize, however, that in Section 3.3 we will obtain dimension-free generalization bounds by imposing additional structure (a Bernstein condition) and working in the PL regime.

## 3.2 LEARNING BOUNDS WITH POLYAK-ŁOJASIEWICZ CONDITION

In non-convex optimization under the Polyak-Łojasiewicz (PL) condition, we are interested in upper bounds on the function-value error. Accordingly, we measure optimization performance and generalization performance via $F_S(\mathbf{x}_{T+1}) - F_S(\mathbf{x}(S))$ and $F(\mathbf{x}_{T+1}) - F(\mathbf{x}^*)$, respectively.

### 3.2.1 CONVERGENCE BOUNDS

We first present high-probability convergence bounds for SGDM under the PL condition.

**Theorem 3.5.** *Let $\mathbf{x}_t$ be the sequence of iterates generated by Algorithm 1. Set the stepsize as $\eta_t = \frac{1}{\mu(S)(t+t_0)}$ such that $t_0 \geq \max\{\frac{12L-4L\gamma}{\mu(S)(1-\gamma)^3}, \frac{(8C_\gamma)L}{(1-\gamma)^2\mu(S)} + 1, \frac{8C_\gamma(L\gamma+L\gamma(C_\gamma))}{(1-\gamma)\mu(S)} - 1, 1\}$, where $C_\gamma = 1 + \frac{2}{\ln^2\gamma} - \frac{3}{\ln\gamma}$ is a constant that depends only on $\gamma$.*

*(1). If $\theta = \frac{1}{2}$, suppose Assumptions 2.1 and 2.9 hold, and assume that $F_S$ satisfies Assumption 2.7 with parameter $2\mu(S)$. Then, for any $\delta \in (0, 1)$, with probability $1 - \delta$,*

$$F_S(\mathbf{x}_{T+1}) - F_S(\mathbf{x}(S)) = \mathcal{O}\left(\frac{\log(1/\delta)}{T}\right).$$

*(2). If $\frac{1}{2} < \theta \leq 1$, suppose Assumptions 2.1, 2.5 and 2.9 hold, and assume that $F_S$ satisfies Assumption 2.7 with parameter $2\mu(S)$. Then, for any $\delta \in (0, 1)$, with probability $1 - \delta$,*

$$F_S(\mathbf{x}_{T+1}) - F_S(\mathbf{x}(S)) = \mathcal{O}\left(\frac{\log^{\theta+\frac{3}{2}}(1/\delta)\log^{1/2}T}{T}\right).$$

*(3). If $\theta > 1$, suppose Assumptions 2.1, 2.5 and 2.9 hold, and assume that $F_S$ satisfies Assumption 2.7 with parameter $2\mu(S)$. Then, for any $\delta \in (0, 1)$, with probability $1 - \delta$, we have the following inequality*

$$F_S(\mathbf{x}_{T+1}) - F_S(\mathbf{x}(S)) = \mathcal{O}\left(\frac{\log^{\theta+\frac{3}{2}}(1/\delta)\log^{\frac{3(\theta-1)}{2}}(T/\delta)\log^{1/2}T}{T}\right).$$

*Remark* 3.6. Theorem 3.5 shows that, under the PL condition, SGDM enjoys fast convergence rates: the $\widetilde{\mathcal{O}}(1/\sqrt{T})$ rate in Theorem 3.1 is improved to a faster $\widetilde{\mathcal{O}}(1/T)$ rate. By the smoothness property in Lemma C.7, $\|\nabla F_S(\mathbf{x}_{T+1})\|^2 \leq 2L(F_S(\mathbf{x}_{T+1}) - F_S(\mathbf{x}(S)))$, so the bounds in Theorem 3.5 also apply (up to constants) to the squared gradient norm $\|\nabla F_S(\mathbf{x}_{T+1})\|^2$. Moreover, when $\theta = 1/2$, Assumption 2.5 is not needed. As in the non-PL case, larger $\theta$ (heavier tails) deteriorates the convergence rate. One can also verify that these PL-based convergence bounds are strictly sharper than the corresponding results in Li & Orabona (2020); Cutkosky & Mehta (2021). To the best of our knowledge, fast $\widetilde{\mathcal{O}}(1/T)$ high-probability rates for SGDM in non-convex settings under PL-type assumptions have not previously been established in the literature.

### 3.2.2 GENERALIZATION BOUNDS

We next present high-probability generalization bounds for SGDM under the PL condition.

**Theorem 3.7.** *Let $\mathbf{x}_t$ be the sequence of iterates generated by Algorithm 1. Set the stepsize as $\eta_t = \frac{1}{\mu(S)(t+t_0)}$ such that $t_0 \geq \max\{\frac{12L-4L\gamma}{\mu(S)(1-\gamma)^3}, \frac{(8C_\gamma)L}{(1-\gamma)^2\mu(S)} + 1, \frac{8C_\gamma(L\gamma+L\gamma(C_\gamma))}{(1-\gamma)\mu(S)} - 1, 1\}$, where $C_\gamma = 1 + \frac{2}{\ln^2\gamma} - \frac{3}{\ln\gamma}$ is a constant that depends only on $\gamma$, and choose $T \asymp n$.*

*(1). If $\theta = \frac{1}{2}$, suppose Assumptions 2.1 and 2.9 hold, assume that $F_S$ satisfies Assumption 2.7 with parameter $2\mu(S)$, and that $F$ satisfies Assumption 2.7 with parameter $2\mu$. Then, for any $\delta \in (0, 1)$, with probability $1 - \delta$,*

$$F(\mathbf{x}_{T+1}) - F(\mathbf{x}^*) = \mathcal{O}\left(\frac{d + \log(1/\delta)}{n}\log^2\left(\frac{1}{\delta}\right)\log n\right).$$

*(2). If $\frac{1}{2} < \theta \leq 1$, suppose Assumptions 2.1, 2.5 and 2.9 hold, and assume that $F_S$ satisfies Assumption 2.7 with parameter $2\mu(S)$, and that $F$ satisfies Assumption 2.7 with parameter $2\mu$. Then, for any $\delta \in (0, 1)$, with probability $1 - \delta$,*

$$F(\mathbf{x}_{T+1}) - F(\mathbf{x}^*) = \mathcal{O}\left(\frac{d + \log(1/\delta)}{n}\log^{2\theta+1}\left(\frac{1}{\delta}\right)\log n\right).$$

*(3). If $\theta > 1$, suppose Assumptions 2.1, 2.5 and 2.9 hold, and assume that $F_S$ satisfies Assumption 2.7 with parameter $2\mu(S)$, and that $F$ satisfies Assumption 2.7 with parameter $2\mu$. Then, for any $\delta \in (0,1)$, with probability $1 - \delta$,*

$$F(\mathbf{x}_{T+1}) - F(\mathbf{x}^*) = \mathcal{O}\left(\frac{d + \log(1/\delta)}{n} \log^{2\theta+1}\left(\frac{1}{\delta}\right) \log^{\frac{3(\theta-1)}{2}}\left(\frac{n}{\delta}\right) \log n\right).$$

*Remark* 3.8. The quantity $F(\mathbf{x}_{T+1}) - F(\mathbf{x}^*)$ measures the gap between the population risk of the last iterate and the optimal population risk, and is often referred to as the *excess risk* in learning theory (Feldman & Vondrak, 2019; Bassily et al., 2020). Theorem 3.7 shows that, when both the empirical risk $F_S$ and population risk $F$ satisfy the PL condition, SGDM enjoys generalization bounds of order $\widetilde{\mathcal{O}}((d + \log(1/\delta))/n)$, improving the dependence on $n$ compared to the general non-convex case in Theorem 3.3. By the smoothness property in Lemma C.7, $\|\nabla F(\mathbf{x}_{T+1})\|^2 \leq 2L(F(\mathbf{x}_{T+1}) - F(\mathbf{x}^*))$, so the bounds in Theorem 3.7 also directly control $\|\nabla F(\mathbf{x}_{T+1})\|^2$. We also emphasize that, in contrast with Section 3.1, the bounds in Section 3.2 are stated for the *last iterate* of SGDM rather than the time-averaged iterate. Overall, the pair of results Theorems 3.5 and 3.7 illustrates the qualitative picture that in the general non-convex regime one can expect $\widetilde{\mathcal{O}}(1/\sqrt{T})$ and $\widetilde{\mathcal{O}}(1/\sqrt{n})$ rates, while under PL-type curvature the rates improve to $\widetilde{\mathcal{O}}(1/T)$ and $\widetilde{\mathcal{O}}(1/n)$, respectively.

### 3.3 Learning Bounds with Bernstein Condition

In this section, we derive sharper generalization bounds by imposing the Bernstein condition.

**Theorem 3.9.** *Assume that the set $\mathcal{X}$ satisfies $\mathcal{X} \subseteq B(\mathbf{x}^*, R)$. Let $\mathbf{x}_t$ be the sequence of iterates generated by Algorithm 1. Set the stepsize as $\eta_t = \frac{1}{\mu(S)(t+t_0)}$ such that $t_0 \geq \max\{\frac{12L-4L\gamma}{\mu(S)(1-\gamma)^3}, \frac{(8C_\gamma)L}{(1-\gamma)^2\mu(S)}+1, \frac{8C_\gamma(L\gamma+L\gamma(C_\gamma))}{(1-\gamma)\mu(S)}-1, 1\}$, where $C_\gamma = 1 + \frac{2}{\ln^2\gamma} - \frac{3}{\ln\gamma}$ is a constant that depends only on $\gamma$, and choose $T \asymp n^2$.*

*(1). If $\theta = \frac{1}{2}$, suppose Assumptions 2.1, 2.3 and 2.9 hold, assume that $F_S$ satisfies Assumption 2.7 with parameter $2\mu(S)$, and that $F$ satisfies Assumption 2.7 with parameter $2\mu$. If $n \geq \frac{cL^2(d+\log(\frac{8\log(2nR+2)}{\delta}))}{\mu^2}$, where $c$ is an absolute constant, then for any $\delta \in (0,1)$, with probability $1 - \delta$,*

$$F(\mathbf{x}_{T+1}) - F(\mathbf{x}^*) = \mathcal{O}\left(\frac{\log^2(1/\delta)}{n^2} + \frac{F(\mathbf{x}^*)\log(1/\delta)}{n}\right).$$

*(2). If $\frac{1}{2} < \theta \leq 1$, suppose Assumptions 2.1, 2.3, 2.5 and 2.9 hold, assume that $F_S$ satisfies Assumption 2.7 with parameter $2\mu(S)$, and that $F$ satisfies Assumption 2.7 with parameter $2\mu$. If $n \geq \frac{cL^2(d+\log(\frac{8\log(2nR+2)}{\delta}))}{\mu^2}$, where $c$ is an absolute constant, then for any $\delta \in (0,1)$, with probability $1 - \delta$,*

$$F(\mathbf{x}_{T+1}) - F(\mathbf{x}^*) = \mathcal{O}\left(\frac{\log^{\theta+\frac{3}{2}}(1/\delta) \log^{1/2} n}{n^2} + \frac{F(\mathbf{x}^*)\log(1/\delta)}{n}\right).$$

*(3). If $\theta > 1$, suppose Assumptions 2.1, 2.3, 2.5 and 2.9 hold, assume that $F_S$ satisfies Assumption 2.7 with parameter $2\mu(S)$, and that $F$ satisfies Assumption 2.7 with parameter $2\mu$. If $n \geq \frac{cL^2(d+\log(\frac{8\log(2nR+2)}{\delta}))}{\mu^2}$, where $c$ is an absolute constant, then for any $\delta \in (0,1)$, with probability $1 - \delta$,*

$$F(\mathbf{x}_{T+1}) - F(\mathbf{x}^*) = \mathcal{O}\left(\frac{\log^{\frac{3(\theta-1)}{2}}(n/\delta) \log^{\theta+\frac{3}{2}}(1/\delta) \log^{1/2} n}{n^2} + \frac{F(\mathbf{x}^*)\log(1/\delta)}{n}\right).$$

*(4). Furthermore, if we additionally assume $F(\mathbf{x}^*) = \mathcal{O}(1/n)$, then the bounds in (1)–(3) simplify, respectively, to*

$$\mathcal{O}\left(\frac{\log^2(1/\delta)}{n^2}\right), \quad \mathcal{O}\left(\frac{\log^{\theta+\frac{3}{2}}(1/\delta) \log^{1/2} n}{n^2}\right), \quad \mathcal{O}\left(\frac{\log^{\frac{3(\theta-1)}{2}}(n/\delta) \log^{\theta+\frac{3}{2}}(1/\delta) \log^{1/2} n}{n^2}\right).$$

*Remark* 3.10. Theorem 3.9 shows that, under the assumptions of Theorem 3.7 together with the Bernstein condition, the excess risk can be improved to

$$\widetilde{\mathcal{O}}\left( \frac{F(\mathbf{x}^*)}{n} + \frac{1}{n^2} \right).$$

Here $F(\mathbf{x}^*)$ is the minimal population risk and is typically very small. Compared with Theorems 3.3 and 3.7, Theorem 3.9 therefore yields strictly sharper bounds. A well-known drawback of the uniform-convergence approach is that, for general non-convex problems, it usually leads to learning bounds with a square-root dependence on the dimension $d$ (Feldman, 2016), as seen in Theorem 3.3. A distinctive advantage of Theorem 3.9 is that, by exploiting Assumption 2.3, we remove the dependence on $d$ in the upper bound, making the bounds more suitable for high-dimensional models. The auxiliary assumption $F(\mathbf{x}^*) = \mathcal{O}(1/n)$ in part (4) is only used to illustrate the attainable rates under a low-noise condition. Strictly speaking, $F(\mathbf{x}^*)$ is independent of $n$, but assumptions such as $F(\mathbf{x}^*) = \mathcal{O}(1/n)$ or even $F(\mathbf{x}^*) = 0$ are standard in the literature; see, for example, Zhang et al. (2017); Srebro et al. (2010); Liu et al. (2018); Lei & Ying (2020). In general, $\mathcal{O}(1/n^2)$-type generalization bounds are rare in learning theory. Theorem 3.9 provides, to the best of our knowledge, the first high-probability $\widetilde{\mathcal{O}}(1/n^2)$ generalization guarantees for SGDM.

## 4 CONCLUSIONS

This paper investigates high-probability convergence and generalization bounds for stochastic gradient descent with momentum (SGDM) in non-convex settings, thus providing a unified view of its optimization and generalization behavior. We hope that these results offer a clearer theoretical picture of when and how SGDM is guaranteed to perform well, and that they serve as a foundation for further studies of momentum-based methods in modern non-convex learning problems.

## ACKNOWLEDGMENTS

This work was supported by the National Key Research and Development Program of China (No. 2024YFE0203200), the National Natural Science Foundation of China (No. 62476277), the CCF-ALIMAMA TECH Kangaroo Fund (No. CCF-ALIMAMA OF 2024008), and the Huawei–Renmin University Joint Program on Information Retrieval. We also acknowledge the support provided by the fund for building worldclass universities (disciplines) of Renmin University of China and by the funds from Beijing Key Laboratory of Big Data Management and Analysis Methods, Gaoling School of Artificial Intelligence, Renmin University of China, from Engineering Research Center of Next Generation Intelligent Search and Recommendation, Ministry of Education, from Intelligent Social Governance Interdisciplinary Platform, Major Innovation & Planning Interdisciplinary Platform for the "DoubleFirst Class" Initiative, Renmin University of China, from Public Policy and Decision making Research Lab of Renmin University of China, and from Public Computing Cloud, Renmin University of China.

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

## A  NUMERICAL EXPERIMENTS

We now present numerical experiments illustrating how the generalization bounds behave as the tail parameter $\theta$ varies. Let $F_S(\mathbf{x})$ and $F_{S'}(\mathbf{x})$ denote the risks built on the training set $S$ and the test set $S'$, respectively, where

$$F_{S'}(\mathbf{x}) = \frac{1}{|S'|} \sum_{z \in S'} f(\mathbf{x}; z),$$

and $|S'|$ is the cardinality of $S'$. We use $F_{S'}(\mathbf{x})$ as an empirical proxy for the population risk $F(\mathbf{x})$.

We consider six datasets available from the LIBSVM dataset: Heart, Fourclass, German, Australian, Diabetes, and Phishing (Chang & Lin, 2011). For each dataset, we take 80 percents as the training dataset and leave the remaining 20 percents as the testing dataset. According to Algorithm 1, the momentum update can be written as

$$\mathbf{m}_t = \gamma \mathbf{m}_{t-1} + \eta_t(\nabla F_S(\mathbf{x}_t) + \nabla f(\mathbf{x}_t; z_{j_t}) - \nabla F_S(\mathbf{x}_t)) = \gamma \mathbf{m}_{t-1} + \eta_t(\nabla F_S(\mathbf{x}_t) + \mathbf{e}_t),$$

where $\mathbf{e}_t = \nabla f(\mathbf{x}_t; z_{j_t}) - \nabla F_S(\mathbf{x}_t)$ is the gradient noise. In each update of our experiments, for each coordinate we independently draw a sample from a sub-Weibull distribution to model $\mathbf{e}_t$ in Assumption 2.9. If every coordinate of $\mathbf{e}_t$ is sub-Weibull, then $|\mathbf{e}_t|$ is also sub-Weibull; this follows from Lemma 3.4 of Bastianello et al. (2021) and part (c) of Proposition 2.1 of Kim et al. (2021b). Since we assume that the stochastic gradient is an unbiased estimator of the exact gradient, we shift and scale the distribution in order to get a random vector with zero mean and the variance equal 1. To examine the effect of the tail parameter, we consider $\theta \in \{1/2, 1, 5\}$.

We work with a generalized linear model $\ell(\langle \mathbf{x}, x \rangle)$ for binary classification, where $\ell$ is the logistic link function $\ell(s) = (1+e^{-s})^{-1}$. Our first experiment uses the Huber loss: $f(\mathbf{x}, z) = \frac{1}{2}(\ell(\langle \mathbf{x}, x \rangle) - y)^2$ if $|\ell(\langle \mathbf{x}, x \rangle) - y| \leq \tau$ and $\tau(|\ell(\langle \mathbf{x}, x \rangle) - y| - \frac{1}{2}\tau)$ otherwise. We set $\tau = 0.1$, $\gamma = 0.9$ and $\eta_t = 0.1t^{-\frac{1}{2}}$, run the algorithm for a given number of passes over the data, repeat experiments 100 times, and report the average of results. The behavior of the empirical quantity $\frac{1}{T}\sum_{t=1}^{T} \|\nabla F_{S'}(\mathbf{x}_t)\|^2$ as a function of the number of passes is shown in Fig. 1. The curves are consistent with the generalization bounds of Theorem 3.3: larger $\theta$ (heavier tails) yield worse generalization behavior, and the case $\theta = 5$ performs noticeably worse, in line with the theoretical regime $\theta > 1$.

Our second experiment uses the squared loss: $f(\mathbf{x}, z) = (\ell(\langle \mathbf{x}, x \rangle) - y)^2$. The corresponding behavior of $\frac{1}{T}\sum_{t=1}^{T} \|\nabla F_{S'}(\mathbf{x}_t)\|^2$ versus the number of passes is reported in Fig. 2. Again, increasing $\theta$ systematically leads to worse generalization performance, which is in clear agreement with Theorem 3.3.

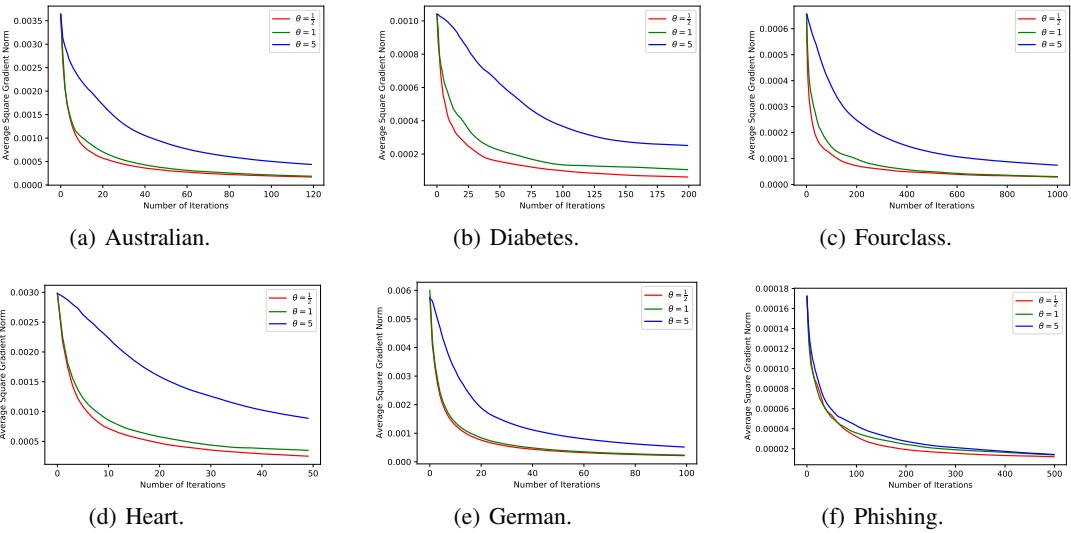

(a) Australian.  (b) Diabetes.  (c) Fourclass.

(d) Heart.  (e) German.  (f) Phishing.

Figure 1: The generalization bound $\frac{1}{T}\sum_{t=1}^{T} \|\nabla F_{S'}(\mathbf{x}_t)\|^2$ versus the number of passes for different choices of $\theta \in \{1/2, 1, 5\}$ and different datasets in the setting of huber loss.

## B  SUMMARY OF RESULTS

We compare the main results of this paper with the most relevant high-probability results of SGDM in the literature in Table 1.

We briefly explain the notation used in Table 1. Entry [1] corresponds to Li & Orabona (2020), and entry [2] to Cutkosky & Mehta (2021). The second result of [1] is derived for a variant of

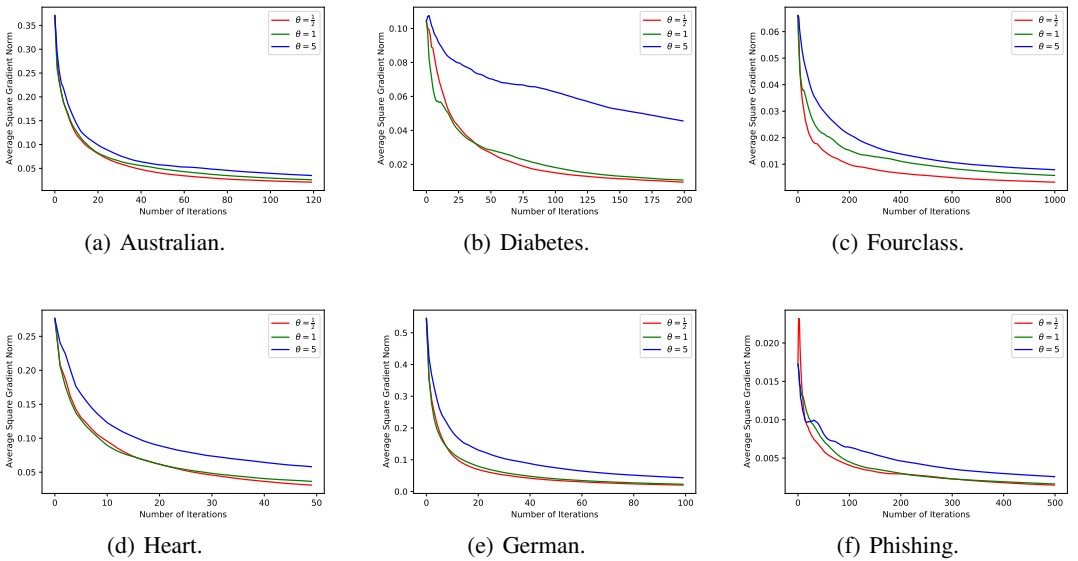

Figure 2: The generalization bound $\frac{1}{T}\sum_{t=1}^{T}\|\nabla F_{S'}(\mathbf{x}_t)\|^2$ versus the number of passes for different choices of $\theta \in \{1/2, 1, 5\}$ and different datasets in the setting of square loss.

SGDM, namely *delayed AdaGrad with momentum*, whose stepsize does not depend on the current gradient. The assumption "$\theta$-order moment" means that the gradient satisfies $\mathbb{E}_z[\|\nabla f(\mathbf{x}_t; z)\|^\theta] \leq G^\theta$ for some constant $G$ and $\theta \in (1, 2]$. "S-S" denotes a second-order smoothness assumption (Cutkosky & Mehta, 2021). Cutkosky & Mehta (2021) also derive two additional convergence bounds (Theorems 3 and 6 therein) for the last iterate of SGDM under a warm-up learning-rate schedule and several other tricks. These bounds have rates similar to those reported for [2] in Table 1, but their assumptions are rather involved and hard to summarize concisely, so we omit them for brevity. "LN" stands for the low-noise condition $F(\mathbf{x}^*) = \mathcal{O}(1/n)$, and the parameter $\theta$ in the table refers to Assumption 2.9.

The detailed comparisons between our bounds and those of Li & Orabona (2020) and Cutkosky & Mehta (2021) have already been discussed in the main text (see the corresponding remarks), so we do not repeat them here. At a glance, Table 1 shows that our work provides a collection of high-probability generalization bounds that are not available in the prior literature, together with convergence bounds that achieve strictly faster rates under comparable assumptions.

# C    PRELIMINARIES

This section collects preliminaries, including basic properties of the sub-Weibull distribution and several auxiliary lemmas used in the proofs.

## C.1    SUB-WEIBULL DISTRIBUTION

Define the $L_p$ norm of a random variable $X$ by $\|X\|_p = (\mathbb{E}|X|^p)^{1/p}$ for any $p \geq 1$. A sub-Weibull random variable $X$ (denoted $X \sim \mathrm{subW}(\theta, K)$) can be characterized in several equivalent ways.

**Proposition C.1** ((Vladimirova et al., 2020; Bastianello et al., 2021)). *Given $\theta \geq 0$, the following properties are equivalent:*

- $\exists K_1 > 0$ *such that* $P(|X| \geq t) \leq 2\exp\left(-(t/K_1)^{1/\theta}\right), \forall t > 0$;

- $\exists K_2 > 0$ *such that* $\|X\|_k \leq K_2 k^\theta, \forall k \geq 1$;

- $\exists K_3 > 0$ *such that* $\mathbb{E}[\exp\left((\lambda|X|)^{1/\theta}\right)] \leq \exp\left((\lambda K_3)^{1/\theta}\right), \forall \lambda \in (0, 1/K_3)$;

Table 1: Summary of Results.

| REF. | ASSUMPTION | MEASURE | LEARNING BOUND |
|---|---|---|---|
| [1] | 2.1, $\theta=\frac{1}{2}$ | $\frac{1}{T}\sum_{t=1}^T \|\nabla F_S(\mathbf{x}_t)\|^2$ | $\mathcal{O}\left(\frac{\log(T/\delta)\log T}{\sqrt{T}}\right)$ |
| | 2.1, $\theta=\frac{1}{2}$ | $\frac{1}{T}\sum_{t=1}^T \|\nabla F_S(\mathbf{x}_t)\|^2$ | $\max\left\{\mathcal{O}\left(\frac{d\log^{\frac{3}{2}}(T/\delta)}{\sqrt{T}}\right),\mathcal{O}\left(\frac{d^2\log^2(T/\delta)}{T}\right)\right\}$ |
| [2] | $\theta$-ORDER MOMENT ($\theta\in(1,2]$), 2.1 | $\frac{1}{T}\sum_{t=1}^T \|\nabla F_S(\mathbf{x}_t)\|$ | $\mathcal{O}\left(\frac{\log(T/\delta)}{T^{\frac{p-1}{3p-2}}}\right)$ |
| | $\theta$-ORDER MOMENT ($\theta\in(1,2]$), 2.1, S-S | $\frac{1}{T}\sum_{t=1}^T \|\nabla F_S(\mathbf{x}_t)\|$ | $\mathcal{O}\left(\frac{\log(T/\delta)}{T^{\frac{2p-2}{5p-3}}}\right)$ |
| OURS | 2.1, $\theta=\frac{1}{2}$ | $\frac{1}{T}\sum_{t=1}^T \|\nabla F_S(\mathbf{x}_t)\|^2$ | $\mathcal{O}\left(\frac{\log(1/\delta)\log T}{\sqrt{T}}\right)$ |
| | 2.1, 2.5, $\theta\in(\frac{1}{2},1]$ | $\frac{1}{T}\sum_{t=1}^T \|\nabla F_S(\mathbf{x}_t)\|^2$ | $\mathcal{O}\left(\frac{\log^{2\theta}(1/\delta)\log T}{\sqrt{T}}\right)$ |
| | 2.1, 2.5, $\theta>1$ | $\frac{1}{T}\sum_{t=1}^T \|\nabla F_S(\mathbf{x}_t)\|^2$ | $\mathcal{O}\left(\frac{\log^{\theta-1}(T/\delta)\log(1/\delta)+\log^{2\theta}(1/\delta)\log T}{\sqrt{T}}\right)$ |
| | 2.1, $\theta=\frac{1}{2}$ | $\frac{1}{T}\sum_{t=1}^T \|\nabla F(\mathbf{x}_t)\|^2$ | $\mathcal{O}\left(\left(\frac{d}{n}\right)^{\frac{1}{2}}\log(\frac{n}{d})\log^3(\frac{1}{\delta})\right)$ |
| | 2.1, 2.5, $\theta\in(\frac{1}{2},1]$ | $\frac{1}{T}\sum_{t=1}^T \|\nabla F(\mathbf{x}_t)\|^2$ | $\mathcal{O}\left(\left(\frac{d}{n}\right)^{\frac{1}{2}}\log(\frac{n}{d})\log^{(2\theta+2)}(\frac{1}{\delta})\right)$ |
| | 2.1, 2.5, $\theta>1$ | $\frac{1}{T}\sum_{t=1}^T \|\nabla F(\mathbf{x}_t)\|^2$ | $\mathcal{O}\left(\left(\frac{d}{n}\right)^{\frac{1}{2}}(\log(\frac{n}{d})\log^{(2\theta+2)}(\frac{1}{\delta})+\log^{\theta-1}(\frac{n}{d\delta})\log^2(\frac{1}{\delta}))\right)$ |
| | 2.1, 2.7, $\theta=\frac{1}{2}$ | $F_S(\mathbf{x}_{T+1})-F_S(\mathbf{x}(S))$ | $\mathcal{O}\left(\frac{\log(1/\delta)}{T}\right)$ |
| | 2.1, 2.5, 2.7, $\theta\in(\frac{1}{2},1]$ | $F_S(\mathbf{x}_{T+1})-F_S(\mathbf{x}(S))$ | $\mathcal{O}\left(\frac{\log^{(\theta+\frac{3}{2})}(\frac{1}{\delta})\log^{\frac{1}{2}}T}{T}\right)$ |
| | 2.1, 2.5, 2.7, $\theta>1$ | $F_S(\mathbf{x}_{T+1})-F_S(\mathbf{x}(S))$ | $\mathcal{O}\left(\frac{\log^{(\theta+\frac{3}{2})}(\frac{1}{\delta})\log^{\frac{3(\theta-1)}{2}}(T/\delta)\log^{\frac{1}{2}}T}{T}\right)$ |
| | 2.1, 2.7, $\theta=\frac{1}{2}$ | $F(\mathbf{x}_{T+1})-F(\mathbf{x}^*)$ | $\mathcal{O}\left(\frac{d+\log(\frac{1}{\delta})}{n}\log^2(\frac{1}{\delta})\log n\right)$ |
| | 2.1, 2.5, 2.7, $\theta\in(\frac{1}{2},1]$ | $F(\mathbf{x}_{T+1})-F(\mathbf{x}^*)$ | $\mathcal{O}\left(\frac{d+\log(\frac{1}{\delta})}{n}\log^{(2\theta+1)}(\frac{1}{\delta})\log n\right)$ |
| | 2.1, 2.5, 2.7, $\theta>1$ | $F(\mathbf{x}_{T+1})-F(\mathbf{x}^*)$ | $\mathcal{O}\left(\frac{d+\log(\frac{1}{\delta})}{n}\log^{(2\theta+1)}(\frac{1}{\delta})\log^{\frac{3(\theta-1)}{2}}(\frac{n}{\delta})\log n\right)$ |
| | 2.1, 2.7, 2.3, $\theta=\frac{1}{2}$ | $F(\mathbf{x}_{T+1})-F(\mathbf{x}^*)$ | $\mathcal{O}\left(\frac{\log^2(\frac{1}{\delta})}{n^2}+\frac{F(\mathbf{x}^*)\log(\frac{1}{\delta})}{n}\right)$ |
| | 2.1, 2.5, 2.7, 2.3, $\theta\in(\frac{1}{2},1]$ | $F(\mathbf{x}_{T+1})-F(\mathbf{x}^*)$ | $\mathcal{O}\left(\frac{\log^{(\theta+\frac{3}{2})}(\frac{1}{\delta})\log^{\frac{1}{2}}n}{n^2}+\frac{F(\mathbf{x}^*)\log(1/\delta)}{n}\right)$ |
| | 2.1, 2.5, 2.7, 2.3, $\theta>1$ | $F(\mathbf{x}_{T+1})-F(\mathbf{x}^*)$ | $\mathcal{O}\left(\frac{\log^{\frac{3(\theta-1)}{2}}(n/\delta)\log^{(\theta+\frac{3}{2})}(\frac{1}{\delta})\log^{\frac{1}{2}}n}{n^2}+\frac{F(\mathbf{x}^*)\log(1/\delta)}{n}\right)$ |
| | 2.1, 2.7, 2.3, LN, $\theta=\frac{1}{2}$ | $F(\mathbf{x}_{T+1})-F(\mathbf{x}^*)$ | $\mathcal{O}\left(\frac{\log^2(\frac{1}{\delta})}{n^2}\right)$ |
| | 2.1, 2.5, 2.7, 2.3, LN, $\theta\in(\frac{1}{2},1]$ | $F(\mathbf{x}_{T+1})-F(\mathbf{x}^*)$ | $\mathcal{O}\left(\frac{\log^{(\theta+\frac{3}{2})}(\frac{1}{\delta})\log^{\frac{1}{2}}n}{n^2}\right)$ |
| | 2.1, 2.5, 2.7, 2.3, LN, $\theta>1$ | $F(\mathbf{x}_{T+1})-F(\mathbf{x}^*)$ | $\mathcal{O}\left(\frac{\log^{\frac{3(\theta-1)}{2}}(n/\delta)\log^{(\theta+\frac{3}{2})}(\frac{1}{\delta})\log^{\frac{1}{2}}n}{n^2}\right)$ |

- $\exists K_4 > 0$ such that $\mathbb{E}\left[\exp\left((|X|/K_4)^{1/\theta}\right)\right] \leq 2$.

*The parameters $K_1, K_2, K_3, K_4$ differ each by a constant that only depends on $\theta$.*

We list several concentration inequalities for sums and martingales with sub-Weibull increments.

**Lemma C.2** ((Vladimirova et al., 2020; Wong et al., 2020; Madden et al., 2024))**.** *Suppose $X_1, \cdots, X_n$ are sub-Weibull($\theta$) random variables with respective parameters $K_1, \ldots, K_n$. Then, for all $t \geq 0$,*

$$P\left(\left|\sum_{i=1}^n X_i\right| \geq t\right) \leq 2\exp\left(-\left(\frac{t}{g(\theta)\sum_{i=1}^n K_i}\right)^{1/\theta}\right),$$

*where $g(\theta) = (4e)^\theta$ for $\theta \leq 1$ and $g(\theta) = 2(2e\theta)^\theta$ for $\theta \geq 1$.*

The next two lemmas provide sub-Weibull analogues of martingale concentration bounds.

**Lemma C.3** (Theorem 2 in (Li, 2021); see also (Fan & Giraudo, 2019))**.** *Let $\theta \in (0, \infty)$ be given. Assume that $(\mathbf{X}_i, i = 1, \cdots, N)$ is a sequence of $\mathbb{R}^d$-valued martingale differences with respect to filtration $\mathcal{F}_i$, i.e. $\mathbb{E}[\mathbf{X}_i|\mathcal{F}_{i-1}] = 0$, and it satisfies the following weak exponential-type tail condition: for some $\theta > 0$ and all $i = 1, ..., N$ we have for some scalar $0 < K_i$, $\mathbb{E}\left[\exp\left(\left\|\frac{\mathbf{X}_i}{K_i}\right\|^{\frac{1}{\theta}}\right)\right] \leq 2$. Assume that $K_i < \infty$ for each $i = 1, ..., N$. Then for an arbitrary $N \geq 1$ and $t > 0$,*

$$P\left(\max_{n\leq N}\left\|\sum_{i=1}^n \mathbf{X}_i\right\| \geq t\right) \leq 4\left[3 + (3\theta)^{2\theta}\frac{128\sum_{i=1}^N K_i^2}{t^2}\right]\exp\left\{-\left(\frac{t^2}{64\sum_{i=1}^N K_i^2}\right)^{\frac{1}{2\theta+1}}\right\}.$$

**Lemma C.4** (Sub-Weibull Freedman Inequality; Proposition 11 in (Madden et al., 2024))**.** *Let $(\Omega, \mathcal{F}, (\mathcal{F}_i), P)$ be a filtered probability space. Let $(\xi_i)$ and $(K_i)$ be adapted to $(\mathcal{F}_i)$. Let $n \in \mathbb{N}$, then for all $i \in [n]$, assume $K_{i-1} \geq 0$, $\mathbb{E}[\xi_i|\mathcal{F}_{i-1}] = 0$, and*

$$\mathbb{E}\left[\exp\left((|\xi_i|/K_{i-1})^{1/\theta}\right)|\mathcal{F}_{i-1}\right] \leq 2$$

*where $\theta \geq 1/2$. If $\theta > 1/2$, assume there exists $(m_i)$ such that $K_{i-1} \leq m_i$.*

*If $\theta = 1/2$, let $a = 2$. Then for all $x, \beta \geq 0$, and $\alpha > 0$, and $\lambda \in \left[0, \frac{1}{2\alpha}\right]$,*

$$P\left(\bigcup_{k\in[n]}\left\{\sum_{i=1}^k \xi_i \geq x \text{ and } \sum_{i=1}^k aK_{i-1}^2 \leq \alpha\sum_{i=1}^k \xi_i + \beta\right\}\right) \leq \exp(-\lambda x + 2\lambda^2\beta). \qquad (2)$$

*and for all $x, \beta, \lambda \geq 0$,*

$$P\left(\bigcup_{k\in[n]}\left\{\sum_{i=1}^k \xi_i \geq x \text{ and } \sum_{i=1}^k aK_{i-1}^2 \leq \beta\right\}\right) \leq \exp\left(-\lambda x + \frac{\lambda^2}{2}\beta\right).$$

*If $\theta \in \left(\frac{1}{2}, 1\right]$, let $a = (4\theta)^{2\theta}e^2$ and $b = (4\theta)^\theta e$. For all $x, \beta \geq 0$, and $\alpha \geq b\max_{i\in[n]} m_i$, and $\lambda \in \left[0, \frac{1}{2\alpha}\right]$,*

$$P\left(\bigcup_{k\in[n]}\left\{\sum_{i=1}^k \xi_i \geq x \text{ and } \sum_{i=1}^k aK_{i-1}^2 \leq \alpha\sum_{i=1}^k \xi_i + \beta\right\}\right) \leq \exp(-\lambda x + 2\lambda^2\beta). \qquad (3)$$

*and for all $x, \beta \geq 0$, and $\lambda \in \left[0, \frac{1}{b\max_{i\in[n]} m_i}\right]$,*

$$P\left(\bigcup_{k\in[n]}\left\{\sum_{i=1}^k \xi_i \geq x \text{ and } \sum_{i=1}^k aK_{i-1}^2 \leq \beta\right\}\right) \leq \exp\left(-\lambda x + \frac{\lambda^2}{2}\beta\right).$$

If $\theta > 1$, let $\delta \in (0,1)$, $a = (2^{2\theta+1} + 2)\Gamma(2\theta + 1) + \frac{2^{3\theta}\Gamma(3\theta+1)}{3}$ and $b = 2\log^{\theta-1}(n/\delta)$. For all $x, \beta \geq 0$, and $\alpha \geq b\max_{i\in[n]} m_i$, and $\lambda \in \left[0, \frac{1}{2\alpha}\right]$,

$$P\left(\bigcup_{k\in[n]} \left\{\sum_{i=1}^{k} \xi_i \geq x \text{ and } \sum_{i=1}^{k} aK_{i-1}^2 \leq \alpha\sum_{i=1}^{k}\xi_i + \beta\right\}\right) \leq \exp(-\lambda x + 2\lambda^2\beta) + 2\delta. \quad (4)$$

and for all $x, \beta \geq 0$, and $\lambda \in \left[0, \frac{1}{b\max_{i\in[n]} m_i}\right]$,

$$P\left(\bigcup_{k\in[n]} \left\{\sum_{i=1}^{k} \xi_i \geq x \text{ and } \sum_{i=1}^{k} aK_{i-1}^2 \leq \beta\right\}\right) \leq \exp\left(-\lambda x + \frac{\lambda^2}{2}\beta\right) + 2\delta.$$

## C.2 AUXILIARY LEMMAS

**Lemma C.5** ((Lei & Tang, 2021)). *Let $e$ be the base of the natural logarithm. There holds the following elementary inequalities.*

*(a) If $\theta \in (0,1)$, then $\sum_{k=1}^{t} k^{-\theta} \leq t^{1-\theta}/(1-\theta)$;*

*(b) If $\theta = 1$, then $\sum_{k=1}^{t} k^{-\theta} \leq \log(et)$;*

*(c) If $\theta > 1$, then $\sum_{k=1}^{t} k^{-\theta} \leq \frac{\theta}{\theta-1}$.*

*(d) $\sum_{k=1}^{t} \frac{1}{k+k_0} \leq \log(t+1)$.*

**Lemma C.6** ((Li & Orabona, 2020)). *For any $T \geq 1$ and sequences $(a_t)$ and $(b_t)$, it holds that*

$$\sum_{t=1}^{T} a_t \sum_{i=1}^{t} b_i = \sum_{t=1}^{T} b_t \sum_{i=t}^{T} a_i \quad \text{and} \quad \sum_{t=1}^{T} a_t \sum_{i=0}^{t-1} b_i = \sum_{t=1}^{T-1} b_t \sum_{i=t+1}^{T} a_i.$$

**Lemma C.7.** *Let $\langle \cdot, \cdot \rangle$ denote the inner product. If $f$ is $L$-smooth, then the following standard properties hold (Nesterov, 2014; Ward et al., 2019): for any $z \in \mathcal{Z}$ and every $\mathbf{x}_1, \mathbf{x}_2$:*

$$f(\mathbf{x}_1; z) - f(\mathbf{x}_2; z) \leq \langle \mathbf{x}_1 - \mathbf{x}_2, \nabla f(\mathbf{x}_2; z) \rangle + \frac{1}{2}L\|\mathbf{x}_1 - \mathbf{x}_2\|^2,$$

$$(2L)^{-1}\|\nabla f(\mathbf{x}; z)\|^2 \leq f(\mathbf{x}; z) - \inf_{\mathbf{x}} f(\mathbf{x}; z).$$

The next two lemmas are uniform-convergence results that control the gap between the population gradient $\nabla F$ and the empirical gradient $\nabla F_S$; they are key tools in our generalization analysis.

**Lemma C.8** (Corollary 2 in (Lei & Tang, 2021)). *Denoted by $B_R = B(\mathbf{0}, R)$. Let $\delta \in (0,1)$ and $S = \{z_1, ..., z_n\}$ be a set of i.i.d. samples. Suppose Assumption 2.1 holds. Then with probability at least $1 - \delta$ we have*

$$\sup_{\mathbf{x}\in B_R} \|\nabla F(\mathbf{x}) - \nabla F_S(\mathbf{x})\| \leq \frac{(LR+B)}{\sqrt{n}}\left(2 + 2\sqrt{48e\sqrt{2}(\log 2 + d\log(3e))} + \sqrt{2\log(\frac{1}{\delta})}\right),$$

*where $B = \sup_{z\in\mathcal{Z}} \|\nabla f(\mathbf{0}; z)\|$ and $L$ is the smoothness constant.*

**Lemma C.9** (Lemma B.4 in (Li & Liu, 2022); (Xu & Zeevi, 2020)). *Suppose Assumptions 2.1 and 2.3 hold, and assume that the population risk $F$ satisfies the PL-type inequality $F(\mathbf{x}) - F(\mathbf{x}^*) \leq \frac{1}{2\mu}\|\nabla F(\mathbf{x})\|^2$ for some $\mu > 0$. If $n \geq \frac{cL^2(d+\log(\frac{8\log(2nR+2)}{\delta}))}{\mu^2}$, then, for all $\mathbf{x} \in \mathcal{X} \subseteq B(\mathbf{x}^*, R)$ and any $\delta > 0$, with probability at least $1 - \delta$*

$$\|\nabla F(\mathbf{x}) - \nabla F_S(\mathbf{x})\| \leq \|\nabla F_S(\mathbf{x})\| + \frac{\mu}{n} + \frac{2B_*\log(4/\delta)}{n} + \sqrt{\frac{8\mathbb{E}[\|\nabla f(\mathbf{x}^*; z)\|^2]\log(4/\delta)}{n}},$$

*where $c$ is an absolute constant, and where $B_*$ is the constant from Assumption 2.3.*

# D PROOF OF MAIN RESULTS

## D.1 PROOF OF THEOREM 3.1

*Proof.* By Assumption 2.1, we have

$$F_S(\mathbf{x}_{t+1}) - F_S(\mathbf{x}_t)$$

$$\leq \langle \mathbf{x}_{t+1} - \mathbf{x}_t, \nabla F_S(\mathbf{x}_t) \rangle + \frac{1}{2} L \|\mathbf{x}_{t+1} - \mathbf{x}_t\|^2 = -\langle \mathbf{m}_t, \nabla F_S(\mathbf{x}_t) \rangle + \frac{1}{2} L \|\mathbf{m}_t\|^2. \tag{5}$$

We first control the term $-\langle \mathbf{m}_t, \nabla F_S(\mathbf{x}_t) \rangle$. We have

$$-\langle \mathbf{m}_t, \nabla F_S(\mathbf{x}_t) \rangle$$

$$= -\gamma \langle \mathbf{m}_{t-1}, \nabla F_S(\mathbf{x}_t) \rangle - \langle \eta_t \nabla f(\mathbf{x}_t; z_{j_t}), \nabla F_S(\mathbf{x}_t) \rangle$$

$$= -\gamma \langle \mathbf{m}_{t-1}, \nabla F_S(\mathbf{x}_{t-1}) \rangle + \gamma \langle \mathbf{m}_{t-1}, \nabla F_S(\mathbf{x}_{t-1}) - \nabla F_S(\mathbf{x}_t) \rangle - \langle \eta_t \nabla f(\mathbf{x}_t; z_{j_t}), \nabla F_S(\mathbf{x}_t) \rangle$$

$$\leq -\gamma \langle \mathbf{m}_{t-1}, \nabla F_S(\mathbf{x}_{t-1}) \rangle - \langle \eta_t \nabla f(\mathbf{x}_t; z_{j_t}), \nabla F_S(\mathbf{x}_t) \rangle + \gamma \|\mathbf{m}_{t-1}\| \|\nabla F_S(\mathbf{x}_{t-1}) - \nabla F_S(\mathbf{x}_t)\|$$

$$\leq -\gamma \langle \mathbf{m}_{t-1}, \nabla F_S(\mathbf{x}_{t-1}) \rangle + L\gamma \|\mathbf{m}_{t-1}\|^2 - \langle \eta_t \nabla f(\mathbf{x}_t; z_{j_t}), \nabla F_S(\mathbf{x}_t) \rangle, \tag{6}$$

where the last inequality uses $L$-smoothness of $F_S$ and the update $\mathbf{x}_t - \mathbf{x}_{t-1} = -\mathbf{m}_{t-1}$. By recurrence and using $\mathbf{m}_0 = 0$, we derive

$$-\langle \mathbf{m}_t, \nabla F_S(\mathbf{x}_t) \rangle \leq L \sum_{i=1}^{t-1} \gamma^{t-i} \|\mathbf{m}_i\|^2 - \sum_{i=1}^{t} \gamma^{t-i} \langle \eta_i \nabla f(\mathbf{x}_i; z_{j_i}), \nabla F_S(\mathbf{x}_i) \rangle. \tag{7}$$

Taking a summation from $t = 1$ to $t = T$ yields

$$F_S(\mathbf{x}_{T+1}) - F_S(\mathbf{x}_1)$$

$$\leq L \sum_{t=1}^{T} \sum_{i=1}^{t-1} \gamma^{t-i} \|\mathbf{m}_i\|^2 - \sum_{t=1}^{T} \sum_{i=1}^{t} \gamma^{t-i} \langle \eta_i \nabla f(\mathbf{x}_i; z_{j_i}), \nabla F_S(\mathbf{x}_i) \rangle + \frac{1}{2} L \sum_{t=1}^{T} \|\mathbf{m}_t\|^2. \tag{8}$$

By Lemma C.6, we have

$$L \sum_{t=1}^{T} \sum_{i=1}^{t-1} \gamma^{t-i} \|\mathbf{m}_i\|^2 \leq L \sum_{t=1}^{T} \gamma^{-t} \|\mathbf{m}_t\|^2 \sum_{i=t}^{T} \gamma^i \leq L \sum_{t=1}^{T} \gamma^{-t} \|\mathbf{m}_t\|^2 \frac{\gamma^t}{1-\gamma} = \frac{L}{1-\gamma} \sum_{t=1}^{T} \|\mathbf{m}_t\|^2. \tag{9}$$

Furthermore, using Lemma C.6, we have

$$-\sum_{t=1}^{T} \sum_{i=1}^{t} \gamma^{t-i} \langle \eta_i \nabla f(\mathbf{x}_t; z_{j_i}), \nabla F_S(\mathbf{x}_i) \rangle$$

$$= -\sum_{t=1}^{T} \sum_{i=1}^{t} \gamma^{t-i} \langle \eta_i (\nabla f(\mathbf{x}_i; z_{j_i}) - \nabla F_S(\mathbf{x}_i)), \nabla F_S(\mathbf{x}_i) \rangle - \sum_{t=1}^{T} \sum_{i=1}^{t} \gamma^{t-i} \langle \eta_i (\nabla F_S(\mathbf{x}_i)), \nabla F_S(\mathbf{x}_i) \rangle$$

$$= -\sum_{t=1}^{T} \gamma^{-t} \langle \eta_t (\nabla f(\mathbf{x}_t; z_{j_t}) - \nabla F_S(\mathbf{x}_t)), \nabla F_S(\mathbf{x}_t) \rangle \sum_{i=t}^{T} \gamma^i - \sum_{t=1}^{T} \gamma^{-t} \langle \eta_t (\nabla F_S(\mathbf{x}_t)), \nabla F_S(\mathbf{x}_t) \rangle \sum_{t=1}^{T} \gamma^i$$

$$\leq -\sum_{t=1}^{T} \gamma^{-t} \langle \eta_t (\nabla f(\mathbf{x}_t; z_{j_t}) - \nabla F_S(\mathbf{x}_t)), \nabla F_S(\mathbf{x}_t) \rangle \sum_{i=t}^{T} \gamma^i - \sum_{t=1}^{T} \eta_t \|\nabla F_S(\mathbf{x}_t)\|^2$$

$$= -\sum_{t=1}^{T} \frac{1-\gamma^{T-t+1}}{1-\gamma} \langle \eta_t (\nabla f(\mathbf{x}_t; z_{j_t}) - \nabla F_S(\mathbf{x}_t)), \nabla F_S(\mathbf{x}_t) \rangle - \sum_{t=1}^{T} \eta_t \|\nabla F_S(\mathbf{x}_t)\|^2. \tag{10}$$

Plugging (9) and (10) into (8), we obtain

$$\sum_{t=1}^{T} \eta_t \|\nabla F_S(\mathbf{x}_t)\|^2 \leq F_S(\mathbf{x}_1) - F_S(\mathbf{x}_S) + \frac{L}{1-\gamma} \sum_{t=1}^{T} \|\mathbf{m}_t\|^2$$

$$-\sum_{t=1}^{T} \frac{1-\gamma^{T-t+1}}{1-\gamma} \langle \eta_t (\nabla f(\mathbf{x}_t; z_{j_t}) - \nabla F_S(\mathbf{x}_t)), \nabla F_S(\mathbf{x}_t) \rangle + \frac{1}{2} L \sum_{t=1}^{T} \|\mathbf{m}_t\|^2. \tag{11}$$

It is clear that

$$\mathbb{E}_{j_t}\left[-\frac{1-\gamma^{T-t+1}}{1-\gamma}\langle\eta_t(\nabla f(\mathbf{x}_t;z_{j_t})-\nabla F_S(\mathbf{x}_t)),\nabla F_S(\mathbf{x}_t)\rangle\right]=0,$$

implying that it is a martingale difference sequence (MDS). We thus use Lemma C.4 to bound it. Specifically, we set $\xi_t = -\frac{1-\gamma^{T-t+1}}{1-\gamma}\langle\eta_t(\nabla f(\mathbf{x}_t;z_{j_t})-\nabla F_S(\mathbf{x}_t)),\nabla F_S(\mathbf{x}_t)\rangle$, $K_{t-1} = \frac{1-\gamma^{T-t+1}}{1-\gamma}\eta_t K\|\nabla F_S(\mathbf{x}_t)\|$, $\beta=0$, $\lambda=\frac{1}{2\alpha}$, and $x=2\alpha\log(1/\delta)$.

If $\theta=\frac{1}{2}$, for all $\alpha>0$, we have the following inequality with probability $1-\delta$

$$-\sum_{t=1}^{T}\frac{1-\gamma^{T-t+1}}{1-\gamma}\langle\eta_t(\nabla f(\mathbf{x}_t;z_{j_t})-\nabla F_S(\mathbf{x}_t)),\nabla F_S(\mathbf{x}_t)\rangle$$

$$\leq 2\alpha\log(1/\delta)+\frac{aK^2}{\alpha}\sum_{t=1}^{T}\eta_t^2(\frac{1-\gamma^{T-t+1}}{1-\gamma})^2\|\nabla F_S(\mathbf{x}_t)\|^2$$

$$\leq 2\alpha\log(1/\delta)+\frac{aK^2}{\alpha}(\frac{1-\gamma^T}{1-\gamma})^2\sum_{t=1}^{T}\eta_t^2\|\nabla F_S(\mathbf{x}_t)\|^2.$$

If $\theta\in(\frac{1}{2},1]$, according to Assumption 2.5, we set $m_t=\frac{1-\gamma^T}{1-\gamma}KG$. Then for all $\alpha\geq b\frac{1-\gamma^T}{1-\gamma}KG$, we have the following inequality with probability $1-\delta$

$$-\sum_{t=1}^{T}\frac{1-\gamma^{T-t+1}}{1-\gamma}\langle\eta_t(\nabla f(\mathbf{x}_t;z_{j_t})-\nabla F_S(\mathbf{x}_t)),\nabla F_S(\mathbf{x}_t)\rangle$$

$$\leq 2\alpha\log(1/\delta)+\frac{aK^2}{\alpha}(\frac{1-\gamma^T}{1-\gamma})^2\sum_{t=1}^{T}\eta_t^2\|\nabla F_S(\mathbf{x}_t)\|^2.$$

If $\theta>1$, according to Assumption 2.5, we set $m_t=\frac{1-\gamma^T}{1-\gamma}KG$ and $\delta=\delta$. Then, for all $\alpha\geq b\frac{1-\gamma^T}{1-\gamma}KG$, we have the following inequality with probability $1-3\delta$

$$-\sum_{t=1}^{T}\frac{1-\gamma^{T-t+1}}{1-\gamma}\langle\eta_t(\nabla f(\mathbf{x}_t;z_{j_t})-\nabla F_S(\mathbf{x}_t)),\nabla F_S(\mathbf{x}_t)\rangle$$

$$\leq 2\alpha\log(1/\delta)+\frac{aK^2}{\alpha}(\frac{1-\gamma^T}{1-\gamma})^2\sum_{t=1}^{T}\eta_t^2\|\nabla F_S(\mathbf{x}_t)\|^2.$$

Then, we control the term $\sum_{t=1}^{T}\|\mathbf{m}_t\|^2$.

$$\sum_{t=1}^{T}\|\mathbf{m}_t\|^2 = \sum_{t=1}^{T}\left\|\gamma\mathbf{m}_{t-1}+(1-\gamma)\frac{\eta_t\nabla f(\mathbf{x}_t;z_{j_t})}{1-\gamma}\right\|^2$$

$$\leq \sum_{t=1}^{T}\left(\gamma\|\mathbf{m}_{t-1}\|^2+(1-\gamma)\left\|\frac{\eta_t\nabla f(\mathbf{x}_t;z_{j_t})}{1-\gamma}\right\|^2\right)$$

$$= \sum_{t=1}^{T-1}\gamma\|\mathbf{m}_t\|^2+\sum_{t=1}^{T}(1-\gamma)\left\|\frac{\eta_t\nabla f(\mathbf{x}_t;z_{j_t})}{1-\gamma}\right\|^2)$$

$$\leq \sum_{t=1}^{T}\gamma\|\mathbf{m}_t\|^2+\sum_{t=1}^{T}(1-\gamma)\left\|\frac{\eta_t\nabla f(\mathbf{x}_t;z_{j_t})}{1-\gamma}\right\|^2),$$

where the first inequality holds due to the Jensen's inequality and the second equality follows from $\|\mathbf{m}_0\|=0$. Thus, we have

$$\sum_{t=1}^{T}\|\mathbf{m}_t\|^2 \leq \sum_{t=1}^{T}\frac{1}{(1-\gamma)^2}\|\eta_t\nabla f(\mathbf{x}_t;z_{j_t})\|^2. \tag{12}$$

This inequality implies that

$$\sum_{t=1}^{T} \|\mathbf{m}_t\|^2 \leq \frac{2}{(1-\gamma)^2} \sum_{t=1}^{T} \eta_t^2 \|\nabla f(\mathbf{x}_t; z_{j_t}) - \nabla F_S(\mathbf{x}_t)\|^2 + \frac{2}{(1-\gamma)^2} \sum_{t=1}^{T} \eta_t^2 \|\nabla F_S(\mathbf{x}_t)\|^2.$$

Since $\|\nabla f(\mathbf{x}_t; z_{j_t}) - \nabla F_S(\mathbf{x}_t)\|$ is a sub-Weibull random variable, we have

$$\mathbb{E}\left[\exp\left(\frac{\eta_t^2 \|\nabla f(\mathbf{x}_t; z_{j_t}) - \nabla F_S(\mathbf{x}_t)\|^2}{\eta_t^2 K^2}\right)^{\frac{1}{2\theta}}\right] \leq 2,$$

which means that $\eta_t^2 \|\nabla f(\mathbf{x}_t; z_{j_t}) - \nabla F_S(\mathbf{x}_t)\|^2 \sim \mathrm{subW}(2\theta, \eta_t^2 K^2)$. Applying Lemma C.2, we get the following inequality with probability $1 - \delta$

$$\sum_{t=1}^{T} \frac{2}{(1-\gamma)^2} \eta_t^2 \|\nabla f(\mathbf{x}_t; z_{j_t}) - \nabla F_S(\mathbf{x}_t)\|^2 \leq \frac{2}{(1-\gamma)^2} K^2 g(2\theta) \log^{2\theta}(2/\delta) \sum_{t=1}^{T} \eta_t^2.$$

Then, we plug the bound of $-\sum_{t=1}^{T} \frac{1-\gamma^{T-t+1}}{1-\gamma} \langle \eta_t (\nabla f(\mathbf{x}_t; z_{j_t}) - \nabla F_S(\mathbf{x}_t)), \nabla F_S(\mathbf{x}_t) \rangle$ and the bound of $\sum_{t=1}^{T} \|\mathbf{m}_t\|^2$ into (11), we obtain

$$\sum_{t=1}^{T} \eta_t \|\nabla F_S(\mathbf{x}_t)\|^2 \leq F_S(\mathbf{x}_1) - F_S(\mathbf{x}(S)) + \left(\frac{L}{1-\gamma} + \frac{1}{2}L\right) \frac{2}{(1-\gamma)^2} \sum_{t=1}^{T} \eta_t^2 \|\nabla F_S(\mathbf{x}_t)\|^2$$

$$+ 2\alpha \log(1/\delta) + \frac{aK^2}{\alpha} \left(\frac{1-\gamma^T}{1-\gamma}\right)^2 \sum_{t=1}^{T} \eta_t^2 \|\nabla F_S(\mathbf{x}_t)\|^2$$

$$+ \left(\frac{L}{1-\gamma} + \frac{1}{2}L\right) \frac{2}{(1-\gamma)^2} K^2 g(2\theta) \log^{2\theta}(2/\delta) \sum_{t=1}^{T} \eta_t^2,$$

implying that

$$\sum_{t=1}^{T} \eta_t \left(1 - \left(\frac{L}{1-\gamma} + \frac{1}{2}L\right) \frac{2}{(1-\gamma)^2} \eta_t - \frac{aK^2}{\alpha} \left(\frac{1-\gamma^T}{1-\gamma}\right)^2 \eta_t\right) \|\nabla F_S(\mathbf{x}_t)\|^2$$

$$\leq F_S(\mathbf{x}_1) - F_S(\mathbf{x}_S) + 2\alpha \log(1/\delta) + \left(\frac{L}{1-\gamma} + \frac{1}{2}L\right) \frac{2}{(1-\gamma)^2} K^2 g(2\theta) \log^{2\theta}(2/\delta) \sum_{t=1}^{T} \eta_t^2.$$

When $c = \eta_1 \leq \frac{1}{8} \frac{(1-\gamma)^2}{\frac{L}{1-\gamma} + \frac{1}{2}L} = \frac{1}{4} \frac{(1-\gamma)^3}{3L - L\gamma}$, then

$$\left(\frac{L}{1-\gamma} + \frac{1}{2}L\right) \frac{2}{(1-\gamma)^2} \eta_t \leq \frac{1}{4}, \forall t. \tag{13}$$

When $\frac{aK^2}{\alpha} \left(\frac{1-\gamma^T}{1-\gamma}\right)^2 \eta_t \leq \frac{1}{4}$, then

$$\alpha \geq 4 \left(\frac{1-\gamma^T}{1-\gamma}\right)^2 \eta_1 aK^2.$$

Thus, if $\alpha \geq 4 \left(\frac{1-\gamma^T}{1-\gamma}\right)^2 \eta_1 aK^2 = 4 \left(\frac{1-\gamma^T}{1-\gamma}\right)^2 caK^2$ and $\eta_1 \leq \frac{1}{8} \frac{(1-\gamma)^2}{\frac{L}{1-\gamma} + \frac{1}{2}L}$, we derive that

$$\sum_{t=1}^{T} \eta_t \|\nabla F_S(\mathbf{x}_t)\|^2$$

$$\leq 2(F_S(\mathbf{x}_1) - F_S(\mathbf{x}(S))) + 4\alpha \log(1/\delta) + 2 \left(\frac{L}{1-\gamma} + \frac{1}{2}L\right) \frac{2}{(1-\gamma)^2} K^2 g(2\theta) \log^{2\theta}(2/\delta) \sum_{t=1}^{T} \eta_t^2.$$

Putting the previous bounds together. Hence, if $\theta = \frac{1}{2}$, taking $\alpha = 4(\frac{1-\gamma^T}{1-\gamma})^2 \eta_1 a K^2 = 8(\frac{1-\gamma^T}{1-\gamma})^2 \eta_1 K^2$, with probability $1 - 2\delta$, we have

$$\sum_{t=1}^{T} \eta_t \|\nabla F_S(\mathbf{x}_t)\|^2 \leq 2(F_S(\mathbf{x}_1) - F_S(\mathbf{x}(S))) + 32(\frac{1-\gamma^T}{1-\gamma})^2 \eta_1 K^2 \log(1/\delta)$$

$$+ (\frac{L}{1-\gamma} + \frac{1}{2}L)\frac{4}{(1-\gamma)^2} K^2 g(1) \log(2/\delta) \sum_{t=1}^{T} \eta_t^2.$$

If $\frac{1}{2} < \theta \leq 1$, taking $\alpha = \max\left\{b\frac{1-\gamma^T}{1-\gamma}KG, 4(\frac{1-\gamma^T}{1-\gamma})^2 \eta_1 a K^2\right\}$
$= \max\left\{(4\theta)^\theta e\frac{1-\gamma^T}{1-\gamma}KG, 4(\frac{1-\gamma^T}{1-\gamma})^2 \eta_1(4\theta)^{2\theta} e^2 K^2\right\}$, with probability $1 - 2\delta$, we have

$$\sum_{t=1}^{T} \eta_t \|\nabla F_S(\mathbf{x}_t)\|^2 \leq 2(F_S(\mathbf{x}_1) - F_S(\mathbf{x}(S)))$$

$$+ 4\max\left\{(4\theta)^\theta e\frac{1-\gamma^T}{1-\gamma}KG, 4(\frac{1-\gamma^T}{1-\gamma})^2 \eta_1(4\theta)^{2\theta} e^2 K^2\right\} \log(\frac{1}{\delta})$$

$$+ (\frac{L}{1-\gamma} + \frac{1}{2}L)\frac{4}{(1-\gamma)^2} K^2 g(2\theta) \log^{2\theta}(2/\delta) \sum_{t=1}^{T} \eta_t^2.$$

If $\theta > 1$, taking $\alpha = \max\left\{b\frac{1-\gamma^T}{1-\gamma}KG, 4(\frac{1-\gamma^T}{1-\gamma})^2 \eta_1 a K^2\right\}$, that is

$$\alpha = \max\left\{2\log^{\theta-1}(T/\delta)\frac{1-\gamma^T}{1-\gamma}KG, 4(\frac{1-\gamma^T}{1-\gamma})^2 \eta_1((2^{2\theta+1}+2)\Gamma(2\theta+1) + \frac{2^{3\theta}\Gamma(3\theta+1)}{3})K^2\right\}.$$

Thus, with probability $1 - 4\delta$, we have

$$\sum_{t=1}^{T} \eta_t \|\nabla F_S(\mathbf{x}_t)\|^2 \leq 2(F_S(\mathbf{x}_1) - F_S(\mathbf{x}(S)))$$

$$+ (\frac{L}{1-\gamma} + \frac{1}{2}L)\frac{4}{(1-\gamma)^2} K^2 g(2\theta) \log^{2\theta}(2/\delta) \sum_{t=1}^{T} \eta_t^2$$

$$+ 4\log(1/\delta)\max\left\{2\log^{\theta-1}(T/\delta)\frac{1-\gamma^T}{1-\gamma}KG,\right.$$

$$\left. 4(\frac{1-\gamma^T}{1-\gamma})^2 \eta_1((2^{2\theta+1}+2)\Gamma(2\theta+1) + \frac{2^{3\theta}\Gamma(3\theta+1)}{3})K^2\right\}.$$

Note that the dependence on confidence parameter $1/\delta$ in above bounds is logarithmic. One can replace $\delta$ to $\delta/2$ or $\delta/4$. Through this simple transformation, we have the following results: (1.) if $\theta = 1$, under Assumptions 2.1 and 2.9, with probability $1 - \delta$, we have

$$\frac{1}{T}\sum_{t=1}^{T} \|\nabla F_S(\mathbf{x}_t)\|^2 \leq \frac{1}{c\sqrt{T}}\sum_{t=1}^{T} \eta_t \|\nabla F_S(\mathbf{x}_t)\|^2 = \mathcal{O}\left(\frac{1}{\sqrt{T}}\log(1/\delta)\sum_{t=1}^{T} \eta_t^2\right)$$

$$= \mathcal{O}\left(\frac{1}{\sqrt{T}}\log(1/\delta)\log T\right); \tag{14}$$

(2.) if $\frac{1}{2} < \theta \leq 1$, under Assumptions 2.1, 2.5, and 2.9, with probability $1 - \delta$, we have

$$\frac{1}{T}\sum_{t=1}^{T} \|\nabla F_S(\mathbf{x}_t)\|^2 \leq \frac{1}{c\sqrt{T}}\sum_{t=1}^{T} \eta_t \|\nabla F_S(\mathbf{x}_t)\|^2 = \mathcal{O}\left(\frac{1}{\sqrt{T}}\log^{2\theta}(1/\delta)\sum_{t=1}^{T} \eta_t^2\right)$$

$$= \mathcal{O}\left(\frac{1}{\sqrt{T}}\log^{2\theta}(1/\delta)\log T\right); \tag{15}$$

(3.) if $\theta > 1$, under Assumptions 2.1, 2.5, and 2.9, with probability $1 - \delta$, we have

$$
\frac{1}{T} \sum_{t=1}^{T} \|\nabla F_S(\mathbf{x}_t)\|^2 \leq \frac{1}{c\sqrt{T}} \sum_{t=1}^{T} \eta_t \|\nabla F_S(\mathbf{x}_t)\|^2
$$

$$
= \mathcal{O} \left( \frac{\log^{\theta-1}(T/\delta) \log(1/\delta) + \log^{2\theta}(1/\delta) \sum_{t=1}^{T} \eta_t^2}{\sqrt{T}} \right)
$$

$$
= \mathcal{O} \left( \frac{\log^{\theta-1}(T/\delta) \log(1/\delta) + \log^{2\theta}(1/\delta) \log T}{\sqrt{T}} \right), \tag{16}
$$

where the bound of $\sum_{t=1}^{T} \eta_t^2$ follows from Lemma C.5. The proof is complete. $\qquad\square$

### D.2 PROOF OF THEOREM 3.3

*Proof.* The proof is divided into three parts.

**(1.)** In the first part, we prove the bound of $\|\mathbf{x}_t\|$. $\|\mathbf{x}_t\|$ characterizes the bound of $B(\mathbf{0}, R)$, i.e., at iterate $t$, $R = R_t = \|\mathbf{x}_t\|$, because $\mathbf{x}_t$ traverses over a ball with an increasing radius as $t$ increases. Therefore one should apply Lemma C.8 with an increasing $R$.

From the update $\mathbf{x}_{t+1} = \mathbf{x}_t - \mathbf{m}_t$, by a summation and using $\mathbf{m}_1 = 0$, we get $\mathbf{x}_{t+1} = -\sum_{i=1}^{t} \mathbf{m}_i$. Using $\mathbf{m}_i = \gamma \mathbf{m}_{i-1} + \eta_i \nabla f(\mathbf{x}_i; z_{j_i})$ and recurrence, we have

$$
\mathbf{m}_i = \sum_{k=1}^{i} \gamma^{i-k} \eta_k \nabla f(\mathbf{x}_k; z_{j_k}).
$$

According to Lemma C.6, this gives that

$$
\mathbf{x}_{t+1} = -\sum_{i=1}^{t} \sum_{k=1}^{i} \gamma^{i-k} \eta_k \nabla f(\mathbf{x}_k; z_{j_k}) = -\sum_{i=1}^{t} \frac{1 - \gamma^{t-i+1}}{1 - \gamma} \eta_i \nabla f(\mathbf{x}_i; z_{j_i}). \tag{17}
$$

Thus, we have

$$
\|\mathbf{x}_{t+1}\| = \frac{1}{1-\gamma} \left\| \sum_{i=1}^{t} (1 - \gamma^{t-i+1}) \eta_i \nabla f(\mathbf{x}_i; z_{j_i}) \right\|
$$

$$
\leq \frac{1}{1-\gamma} \left\| \sum_{i=1}^{t} (1 - \gamma^{t-i+1}) \eta_i (\nabla f(\mathbf{x}_i; z_{j_i}) - \nabla F_S(\mathbf{x}_i)) \right\| + \frac{1}{1-\gamma} \left\| \sum_{i=1}^{t} (1 - \gamma^{t-i+1}) \eta_i \nabla F_S(\mathbf{x}_i) \right\|. \tag{18}
$$

Let's consider the first term $\left\| \sum_{i=1}^{t} (1 - \gamma^{t-i+1}) \eta_i (\nabla f(\mathbf{x}_i; z_{j_i}) - \nabla F_S(\mathbf{x}_i)) \right\|$. It is clear that $\mathbb{E}_{j_i}[(1 - \gamma^{t-i+1}) \eta_i (\nabla f(\mathbf{x}_i; z_{j_i}) - \nabla F_S(\mathbf{x}_i))] = 0$, which means that it is a MDS. Moreover, since $\|\nabla f(\mathbf{x}_i; z_{j_i}) - \nabla F_S(\mathbf{x}_i)\| \sim \mathrm{subW}(\theta, K)$, we have

$$
\mathbb{E} \left[ \exp \left( \frac{\|\eta_i (1 - \gamma^{t-i+1})(\nabla f(\mathbf{x}_i; z_{j_i}) - \nabla F_S(\mathbf{x}_i))\|}{\eta_i (1 - \gamma^t) K} \right)^{\frac{1}{\theta}} \right] \leq 2.
$$

Then, we can apply Lemma C.3 to derive the following inequality

$$
P \left( \max_{1 \leq t \leq T} \left\| \sum_{i=1}^{t} (1 - \gamma^{t-i+1}) \eta_i (\nabla f(\mathbf{x}_i; z_{j_i}) - \nabla F_S(\mathbf{x}_i)) \right\| \geq x \right)
$$

$$
\leq 4 \left[ 3 + (3\theta)^{2\theta} \frac{128 K^2 (1 - \gamma^T) \sum_{i=1}^{T} \eta_i^2}{x^2} \right] \exp \left\{ - \left( \frac{x^2}{64 K^2 (1 - \gamma^T) \sum_{i=1}^{T} \eta_i^2} \right)^{\frac{1}{2\theta+1}} \right\}.
$$

Setting the term $4 \exp\left\{-\left(\frac{x^2}{64K^2(1-\gamma^T)\sum_{i=1}^T \eta_i^2}\right)^{\frac{1}{2\theta+1}}\right\}$ equal to $\delta$, we get $x = 8 \log^{(\theta+\frac{1}{2})}(\frac{4}{\delta})K(1-\gamma^T)^{\frac{1}{2}}(\sum_{i=1}^T \eta_i^2)^{\frac{1}{2}}$. Thus, with probability $1 - 3\delta - \frac{8(3\theta)^{2\theta}}{\log^{2\theta+1}\frac{4}{\delta}}\delta$, we have

$$\max_{1 \le t \le T}\left\|\sum_{i=1}^t (1-\gamma^{t-i+1})\eta_i(\nabla f(\mathbf{w}_i; z_{j_i}) - \nabla F_S(\mathbf{w}_i))\right\| \le 8 \log^{(\theta+\frac{1}{2})}(\frac{4}{\delta})K(1-\gamma^T)^{\frac{1}{2}}\left(\sum_{i=1}^T \eta_i^2\right)^{\frac{1}{2}}. \tag{19}$$

Since $\theta \ge 1/2$ and $\delta \in (0,1)$, we have $\log^{2\theta+1}\frac{4}{\delta} > 1$. Thus, (19) means that with probability $1 - 3\delta - 8(3\theta)^{2\theta}\delta$, we have

$$\max_{1 \le t \le T}\left\|\sum_{i=1}^t (1-\gamma^{t-i+1})\eta_i(\nabla f(\mathbf{w}_i; z_{j_i}) - \nabla F_S(\mathbf{w}_i))\right\| \le 8 \log^{(\theta+\frac{1}{2})}(\frac{4}{\delta})K(1-\gamma^T)^{\frac{1}{2}}\left(\sum_{i=1}^T \eta_i^2\right)^{\frac{1}{2}}.$$

Now, with probability $1 - \delta$, we can derive

$$\max_{1 \le t \le T}\left\|\sum_{i=1}^t (1-\gamma^{t-i+1})\eta_i(\nabla f(\mathbf{w}_i; z_{j_i}) - \nabla F_S(\mathbf{w}_i))\right\|$$
$$\le 8 \log^{(\theta+\frac{1}{2})}\left(\frac{4(3 + 8(3\theta)^{2\theta})}{\delta}\right)K(1-\gamma^T)^{\frac{1}{2}}\left(\sum_{i=1}^T \eta_i^2\right)^{\frac{1}{2}} = \mathcal{O}(1). \tag{20}$$

For the second term $\left\|\sum_{i=1}^t (1-\gamma^{t-i+1})\eta_i \nabla F_S(\mathbf{x}_i)\right\|$, we have

$$\left\|\sum_{i=1}^t (1-\gamma^{t-i+1})\eta_i \nabla F_S(\mathbf{x}_i)\right\|^2 \le \left(\sum_{i=1}^t (1-\gamma^{t-i+1})\eta_i\right)\left(\sum_{i=1}^t (1-\gamma^{t-i+1})\eta_i\|\nabla F_S(\mathbf{x}_i)\|^2\right)$$
$$\le \left(\sum_{i=1}^t \eta_i\right)\left(\sum_{i=1}^t \eta_i\|\nabla F_S(\mathbf{x}_i)\|^2\right), \tag{21}$$

where the first inequality follows form the Schwarz's inequality, and where the second inequality follows from the fact that $0 < \gamma < 1$, $\eta_i > 0$ and $\|\nabla F_S(\mathbf{x}_i)\| \ge 0$. For the sake of the presentation, we introduce a notation $\Delta(\theta, T, \delta) = \log^{\theta-1}(T/\delta)\log(1/\delta)\mathbb{I}_{\theta>1}$, where $\mathbb{I}_{\theta>1}$ is an indication function. Thus with probability $1 - \delta$ we have the following inequality uniformly for all $t = 1, ..., T$

$$\left\|\sum_{i=1}^t (1-\gamma^{t-i+1})\eta_i \nabla F_S(\mathbf{x}_i)\right\|^2 \le \left(\sum_{i=1}^t \eta_i\right)\left(\sum_{i=1}^t \eta_i\|\nabla F_S(\mathbf{x}_i)\|^2\right)$$
$$= \left(\sum_{i=1}^t \eta_i\right)\mathcal{O}\left(\Delta(\theta, T, \delta) + \log^{2\theta}(1/\delta)\sum_{i=1}^t \eta_i^2\right), \tag{22}$$

where the last equation follows from the results of (14), (15), and (16).

Plugging (20), (21) and (22) into (18), we have the following inequality uniformly for all $t = 1, ..., T$ with probability at least $1 - 2\delta$

$$\|\mathbf{x}_{t+1}\| = \mathcal{O}\left(\log^{(\theta+\frac{1}{2})}(\frac{1}{\delta})(1-\gamma^T)^{\frac{1}{2}}(\sum_{i=1}^T \eta_i^2)^{\frac{1}{2}}\right) + \left(\left(\sum_{i=1}^t \eta_i\right)\mathcal{O}\left(\Delta(\theta, T, \delta) + \log^{2\theta}(1/\delta)\sum_{i=1}^t \eta_i^2\right)\right)^{\frac{1}{2}} \tag{23}$$

$$= \mathcal{O}\left(\log^{(\theta+\frac{1}{2})}(\frac{1}{\delta})(1-\gamma^T)^{\frac{1}{2}}\log^{\frac{1}{2}}T\right) + \left(t^{\frac{1}{2}}\mathcal{O}\left(\Delta(\theta, T, \delta) + \log^{2\theta}(1/\delta)\log t\right)\right)^{\frac{1}{2}}$$

$$\le \mathcal{O}\left(t^{\frac{1}{4}}\left(\Delta^{\frac{1}{2}}(\theta, T, \delta) + \log^{(\theta+\frac{1}{2})}(\frac{1}{\delta})\log^{\frac{1}{2}}T\right)\right), \tag{24}$$

where the second equation follows from Lemma C.5.

**(2.)** In the second part, we prove the bound of $\max_{1\le t\le T}\|\nabla F(\mathbf{x}_t)-\nabla F_S(\mathbf{x}_t)\|$. According to Lemma C.8, with probability $1-\delta$ we have

$$\max_{1\le t\le T}\|\nabla F(\mathbf{x}_t)-\nabla F_S(\mathbf{x}_t)\|$$

$$\le\frac{(LR_T+B)}{\sqrt{n}}\left(2+2\sqrt{48e\sqrt{2}(\log 2+d\log(3e))}+\sqrt{2\log(\tfrac{1}{\delta})}\right)$$

$$\le\frac{(L\|\mathbf{x}_T\|+B)}{\sqrt{n}}\left(2+2\sqrt{48e\sqrt{2}(\log 2+d\log(3e))}+\sqrt{2\log(\tfrac{1}{\delta})}\right). \tag{25}$$

Plugging (24) into (25), with probability $1-3\delta$ we have the following inequality uniformly for all $t=1,...T$

$$\max_{1\le t\le T}\|\nabla F(\mathbf{x}_t)-\nabla F_S(\mathbf{x}_t)\|\le$$

$$\frac{L\mathcal{O}\big(T^{\frac{1}{4}}\big(\Delta^{\frac{1}{2}}(\theta,T,\delta)+\log^{(\theta+\frac{1}{2})}(\tfrac{1}{\delta})\log^{\frac{1}{2}}T\big)\big)+B}{\sqrt{n}}\left(2+2\sqrt{48e\sqrt{2}(\log 2+d\log(3e))}+\sqrt{2\log(\tfrac{1}{\delta})}\right),$$

which means that we have the following inequality uniformly for all $t=1,...T$ with probability $1-\delta$

$$\max_{1\le t\le T}\|\nabla F(\mathbf{x}_t)-\nabla F_S(\mathbf{x}_t)\|^2$$

$$=\mathcal{O}\left(\frac{T^{\frac{1}{2}}\big(\Delta(\theta,T,\delta)+\log^{(2\theta+1)}(\tfrac{1}{\delta})\log T\big)}{n}\times\Big(d+\log(\tfrac{1}{\delta})\Big)\right). \tag{26}$$

**(3.)** In the third part, we prove the bound of $\frac{1}{T}\sum_{t=1}^{T}\|\nabla F(\mathbf{x}_t)\|^2$. Firstly, we can derive the following inequality with probability $1-2\delta$

$$\sum_{t=1}^{T}\eta_t\|\nabla F(\mathbf{x}_t)\|^2$$

$$\le 2\sum_{t=1}^{T}\eta_t\|\nabla F(\mathbf{x}_t)-\nabla F_S(\mathbf{x}_t)\|^2+2\sum_{t=1}^{T}\eta_t\|\nabla F_S(\mathbf{x}_t)\|^2$$

$$\le 2\sum_{t=1}^{T}\eta_t\max_{1\le t\le T}\|\nabla F(\mathbf{x}_t)-\nabla F_S(\mathbf{x}_t)\|^2+2\sum_{t=1}^{T}\eta_t\|\nabla F_S(\mathbf{x}_t)\|^2$$

$$\le 2\sum_{t=1}^{T}\eta_t\mathcal{O}\Big(\frac{T^{\frac{1}{2}}\big(\Delta(\theta,T,\delta)+\log^{(2\theta+1)}(\tfrac{1}{\delta})\log T\big)}{n}\Big(d+\log(\tfrac{1}{\delta})\Big)\Big)$$

$$+\mathcal{O}\Big(\Delta(\theta,T,\delta)+\log^{2\theta}(1/\delta)\log T\Big),$$

where the last inequality follows from (26) and the results of (14), (15), and (16).

Therefore, we have

$$\frac{1}{T}\sum_{t=1}^{T}\|\nabla F(\mathbf{x}_t)\|^2\le\frac{1}{c\sqrt{T}}\sum_{t=1}^{T}\eta_t\|\nabla F(\mathbf{x}_t)\|^2$$

$$=\mathcal{O}\left(\frac{\sqrt{T}\big(\Delta(\theta,T,\delta)+\log^{(2\theta+1)}(\tfrac{1}{\delta})\log T\big)}{n}\times\Big(d+\log(\tfrac{1}{\delta})\Big)\right)$$

$$+\mathcal{O}\left(\frac{\Delta(\theta,T,\delta)+\log^{2\theta}(1/\delta)\log T}{\sqrt{T}}\right).$$

Taking $T\asymp\frac{n}{d}$, we have the following inequality with probability $1-2\delta$

$$\frac{1}{T}\sum_{t=1}^{T}\|\nabla F(\mathbf{x}_t)\|^2=\mathcal{O}\left(\Big(\frac{d}{n}\Big)^{\frac{1}{2}}\Big(\log(\tfrac{n}{d})\log^{(2\theta+2)}(\tfrac{1}{\delta})+\Delta(\theta,\tfrac{n}{d},\delta)\log(1/\delta)\Big)\right),$$

which means with probability at least $1 - \delta$ we have

$$\frac{1}{T}\sum_{t=1}^{T}\|\nabla F(\mathbf{x}_t)\|^2 = \mathcal{O}\left(\left(\frac{d}{n}\right)^{\frac{1}{2}}\left(\log(\frac{n}{d})\log^{(2\theta+2)}(\frac{1}{\delta}) + \Delta(\theta,\frac{n}{d},\delta)\log(1/\delta)\right)\right)$$

$$= \mathcal{O}\left(\left(\frac{d}{n}\right)^{\frac{1}{2}}\left(\log(\frac{n}{d})\log^{(2\theta+2)}(\frac{1}{\delta}) + \log^{\theta-1}(n/d\delta)\log^2(1/\delta)\mathbb{I}_{\theta>1}\right)\right).$$

The proof is complete. $\qquad\square$

### D.3 PROOF OF THEOREM 3.5

*Proof.* The proof of Theorem 3.5 is relatively complex and is divided into two parts.

**(1.)** In the first part, we prove the bound of $\|\mathbf{x}_{t+1}\|$, characterizing the bound of $B(\mathbf{0}, R)$, i.e., at iterate $t + 1$, $R = R_{t+1} = \|\mathbf{x}_{t+1}\|$. Recall that in (13), we need $\eta_t \leq \frac{1}{8}\frac{(1-\gamma)^2}{\frac{L}{1-\gamma}+\frac{1}{2}L}$. Since $\eta_t = \frac{1}{\mu(S)(t+t_0)}$, when $t_0 \geq \frac{8(\frac{L}{1-\gamma}+\frac{1}{2}L)}{\mu(S)(1-\gamma)^2} = \frac{12L-4L\gamma}{\mu(S)(1-\gamma)^3}$, we have $\eta_t \leq \frac{1}{8}\frac{(1-\gamma)^2}{\frac{L}{1-\gamma}+\frac{1}{2}L}$. Thus, we can use (23) to bound $\|\mathbf{x}_{t+1}\|$. According to (23), we have the following inequality with probability $1 - \delta$ uniformly for all $t = 1, \dots T$

$$\|\mathbf{x}_{t+1}\| = \mathcal{O}\left(\log^{(\theta+\frac{1}{2})}(\frac{1}{\delta})(\sum_{t=1}^{T}\eta_t^2)^{\frac{1}{2}} + \left(\sum_{i=1}^{t}\eta_i\right)^{\frac{1}{2}}\left(\Delta^{\frac{1}{2}}(\theta,T,\delta) + \log^{\theta}(1/\delta)\left(\sum_{i=1}^{t}\eta_i^2\right)^{\frac{1}{2}}\right)\right)$$

$$\leq \mathcal{O}\left(\left(\log^{(\theta+\frac{1}{2})}(\frac{1}{\delta}) + \Delta^{\frac{1}{2}}(\theta,T,\delta)\right)\log^{\frac{1}{2}}T\right), \tag{27}$$

where $\Delta(\theta,T,\delta) = \log^{\theta-1}(T/\delta)\log(1/\delta)\mathbb{I}_{\theta>1}$, and where the last inequality follows from $\eta_t = \frac{1}{\mu(S)(t+t_0)}$ with $t_0 \geq 1$ and Lemma C.5.

**(2.)** In the second part, we prove the bound of $F_S(\mathbf{x}_{T+1}) - F_S(\mathbf{x}(S))$. It is clear that

$$F_S(\mathbf{x}_{t+1}) - F_S(\mathbf{x}_t)$$

$$\leq \langle \mathbf{x}_{t+1} - \mathbf{x}_t, \nabla F_S(\mathbf{x}_t)\rangle + \frac{1}{2}L\|\mathbf{x}_{t+1} - \mathbf{x}_t\|^2$$

$$\leq -\gamma\langle \mathbf{m}_{t-1}, \nabla F_S(\mathbf{x}_{t-1})\rangle + L\gamma\|\mathbf{m}_{t-1}\|^2 - \langle \eta_t\nabla f(\mathbf{x}_t;z_{j_t}), \nabla F_S(\mathbf{x}_t)\rangle + \frac{1}{2}L\|\mathbf{m}_t\|^2$$

$$= -\gamma\langle \mathbf{m}_{t-1}, \nabla F_S(\mathbf{x}_{t-1})\rangle + L\gamma\|\mathbf{m}_{t-1}\|^2 - \langle \eta_t\nabla f(\mathbf{x}_t;z_{j_t}) - \nabla F_S(\mathbf{x}_t), \nabla F_S(\mathbf{x}_t)\rangle$$

$$- \eta_t\|\nabla F_S(\mathbf{x}_t)\|^2 + \frac{1}{2}L\|\mathbf{m}_t\|^2,$$

where the second inequality follows from (6). We can derive that

$$\frac{1}{2}\eta_t\|\nabla F_S(\mathbf{x}_t)\|^2 + F_S(\mathbf{x}_{t+1}) - F_S(\mathbf{x}_t)$$

$$\leq -\gamma\langle \mathbf{m}_{t-1}, \nabla F_S(\mathbf{x}_{t-1})\rangle + L\gamma\|\mathbf{m}_{t-1}\|^2 - \langle \eta_t\nabla f(\mathbf{x}_t;z_{j_t}) - \nabla F_S(\mathbf{x}_t), \nabla F_S(\mathbf{x}_t)\rangle$$

$$- \frac{1}{2}\eta_t\|\nabla F_S(\mathbf{x}_t)\|^2 + \frac{1}{2}L\|\mathbf{m}_t\|^2.$$

Since $\eta_t = \frac{1}{\mu(S)(t+t_0)}$, it implies that

$$\frac{1}{2}\eta_t\|\nabla F_S(\mathbf{x}_t)\|^2 + F_S(\mathbf{x}_{t+1}) - F_S(\mathbf{x}_S)$$

$$\leq (1 - \frac{2}{t+t_0})(F_S(\mathbf{x}_t) - F_S(\mathbf{x}_S)) - \gamma\langle \mathbf{m}_{t-1}, \nabla F_S(\mathbf{x}_{t-1})\rangle + L\gamma\|\mathbf{m}_{t-1}\|^2$$

$$- \langle \eta_t\nabla f(\mathbf{x}_t;z_{j_t}) - \nabla F_S(\mathbf{x}_t), \nabla F_S(\mathbf{x}_t)\rangle + \frac{1}{2}L\|\mathbf{m}_t\|^2.$$

Multiplying both sides by $(t + t_0)(t + t_0 - 1)$, we get

$$\frac{(t + t_0 - 1)}{2\mu(S)}\|\nabla F_S(\mathbf{x}_t)\|^2 + (t + t_0)(t + t_0 - 1)(F_S(\mathbf{x}_{t+1}) - F_S(\mathbf{x}_S))$$

$$\leq - (t + t_0)(t + t_0 - 1)\gamma\langle\mathbf{m}_{t-1}, \nabla F_S(\mathbf{x}_{t-1})\rangle + (t + t_0)(t + t_0 - 1)L\gamma\|\mathbf{m}_{t-1}\|^2$$

$$+ (t + t_0)(t + t_0 - 1)\frac{1}{2}L\|\mathbf{m}_t\|^2$$

$$+ (t + t_0 - 1)(t + t_0 - 2)(F_S(\mathbf{x}_t) - F_S(\mathbf{x}_S))$$

$$- (t + t_0)(t + t_0 - 1)\eta_t\langle\nabla f(\mathbf{x}_t; z_{j_t}) - \nabla F_S(\mathbf{x}_t), \nabla F_S(\mathbf{x}_t)\rangle.$$

Taking a summation from $t = 1$ to $t = T$, we derive that

$$\sum_{t=1}^{T}\frac{(t + t_0 - 1)}{2\mu(S)}\|\nabla F_S(\mathbf{x}_t)\|^2 + (T + t_0)(T + t_0 - 1)(F_S(\mathbf{x}_{T+1}) - F_S(\mathbf{x}_S))$$

$$\leq -\sum_{t=1}^{T}(t + t_0)(t + t_0 - 1)\gamma\langle\mathbf{m}_{t-1}, \nabla F_S(\mathbf{x}_{t-1})\rangle + \sum_{t=1}^{T}(t + t_0)(t + t_0 - 1)L\gamma\|\mathbf{m}_{t-1}\|^2$$

$$+ \sum_{t=1}^{T}(t + t_0)(t + t_0 - 1)\frac{1}{2}L\|\mathbf{m}_t\|^2$$

$$+ (t_0 - 1)(t_0 - 2)(F_S(\mathbf{x}_1) - F_S(\mathbf{x}_S))$$

$$- \sum_{t=1}^{T}(t + t_0)(t + t_0 - 1)\eta_t\langle\nabla f(\mathbf{x}_t; z_{j_t}) - \nabla F_S(\mathbf{x}_t), \nabla F_S(\mathbf{x}_t)\rangle.$$

Since $\mathbf{m}_0 = 0$, we get

$$\sum_{t=1}^{T}\frac{(t + t_0 - 1)}{2\mu(S)}\|\nabla F_S(\mathbf{x}_t)\|^2 + (T + t_0)(T + t_0 - 1)(F_S(\mathbf{x}_{T+1}) - F_S(\mathbf{x}_S))$$

$$\leq -\sum_{t=1}^{T}(t + t_0)(t + t_0 - 1)\gamma\langle\mathbf{m}_{t-1}, \nabla F_S(\mathbf{x}_{t-1})\rangle + \sum_{t=1}^{T-1}(t + t_0 + 1)(t + t_0)L\gamma\|\mathbf{m}_t\|^2$$

$$+ \sum_{t=1}^{T}(t + t_0)(t + t_0 - 1)\frac{1}{2}L\|\mathbf{m}_t\|^2$$

$$+ (t_0 - 1)(t_0 - 2)(F_S(\mathbf{x}_1) - F_S(\mathbf{x}_S))$$

$$- \sum_{t=1}^{T}(t + t_0)(t + t_0 - 1)\eta_t\langle\nabla f(\mathbf{x}_t; z_{j_t}) - \nabla F_S(\mathbf{x}_t), \nabla F_S(\mathbf{x}_t)\rangle. \tag{28}$$

We first bound the term $\sum_{t=1}^{T-1}(t + t_0 + 1)(t + t_0)\|\mathbf{m}_t\|^2$. Note that from the Jensen's inequality, we have

$$\|\mathbf{m}_t\|^2 = \|\gamma\mathbf{m}_{t-1} + \frac{1 - \gamma}{1 - \gamma}\eta_t\nabla f(\mathbf{x}_t; z_{j_t})\|^2 \leq \gamma\|\mathbf{m}_{t-1}\|^2 + \frac{1}{1 - \gamma}\|\eta_t\nabla f(\mathbf{x}_t; z_{j_t})\|^2.$$

By recurrence, it gives that

$$\|\mathbf{m}_t\|^2 \leq \sum_{i=1}^{t}\frac{\gamma^{t-i}}{1 - \gamma}\|\eta_i\nabla f(\mathbf{x}_i; z_{j_i})\|^2.$$

Thus, we have

$$\sum_{t=1}^{T-1}(t+t_0+1)(t+t_0)\|\mathbf{m}_t\|^2$$

$$\leq \sum_{t=1}^{T-1}(t+t_0+1)(t+t_0)\sum_{i=1}^{t}\frac{\gamma^{t-i}}{1-\gamma}\|\eta_i\nabla f(\mathbf{x}_i;z_{j_i})\|^2$$

$$=\sum_{t=1}^{T-1}\frac{\gamma^{-t}}{1-\gamma}\|\eta_t\nabla f(\mathbf{x}_t;z_{j_t})\|^2\sum_{i=t}^{T-1}\gamma^i(i+t_0+1)(i+t_0) \qquad (29)$$

Considering $\sum_{i=t}^{T-1}(i+t_0+1)(i+t_0)\gamma^i$, we have

$$\sum_{i=t}^{T-1}(i+t_0+1)(i+t_0)\gamma^i$$

$$\leq \int_t^{T-1}(i+t_0+1)(i+t_0)\gamma^i di$$

$$\leq \int_t^{T-1}(i+t_0+1)^2\gamma^i di$$

$$=\frac{\gamma^i}{\ln\gamma}(i+t_0+1)^2\Big|_{i=t}^{i=T-1}-2\int_t^{T-1}(i+t_0+1)\gamma^i di$$

$$=\frac{\gamma^i}{\ln\gamma}(i+t_0+1)^2\Big|_{i=t}^{i=T-1}-2\Big[\frac{\gamma^i}{\ln^2\gamma}(i+t_0+1)\Big|_{i=t}^{i=T-1}-\int_t^{T-1}\gamma^i di\Big].$$

Solving the above integral, and since $\ln\gamma < 0$, we get

$$\sum_{i=t}^{T-1}(i+t_0+1)(i+t_0)\gamma^i$$

$$\leq -\frac{\gamma^t}{\ln\gamma}(t+t_0+1)^2+2\frac{\gamma^t}{\ln^2\gamma}(t+t_0+1)-2\frac{\gamma^t}{\ln\gamma}\leq (C_\gamma)\gamma^t(t+t_0+1)^2, \qquad (30)$$

where $C_\gamma = 1+2\frac{1}{\ln^2\gamma}-\frac{3}{\ln\gamma}$, which is a constant only depend on $\gamma$. Thus, according to (29), we have

$$\sum_{t=1}^{T-1}(t+t_0+1)(t+t_0)\|\mathbf{m}_t\|^2 \leq \sum_{t=1}^{T-1}(t+t_0+1)^2\frac{(C_\gamma)}{(1-\gamma)}\|\eta_t\nabla f(\mathbf{x}_t;z_{j_t})\|^2$$

$$\leq \frac{(C_\gamma)}{(1-\gamma)\mu(S)^2}\sum_{t=1}^{T-1}\frac{(t+t_0+1)^2}{(t+t_0)^2}\|\nabla f(\mathbf{x}_t;z_{j_t})\|^2.$$

And since $\frac{(t+t_0+1)^2}{(t+t_0)^2}=(1+\frac{1}{t+t_0})^2\leq 4$, then we have

$$\sum_{t=1}^{T-1}(t+t_0+1)(t+t_0)\|\mathbf{m}_t\|^2$$

$$\leq \frac{(4C_\gamma)}{(1-\gamma)\mu(S)^2}\sum_{t=1}^{T-1}\|\nabla f(\mathbf{x}_t;z_{j_t})\|^2$$

$$\leq \frac{(8C_\gamma)}{(1-\gamma)\mu(S)^2}\Big(\sum_{t=1}^{T-1}\|\nabla f(\mathbf{x}_t;z_{j_t})-\nabla F_S(\mathbf{x}_t)\|^2+\|\nabla F_S(\mathbf{x}_t)\|^2\Big).$$

Since $\|\nabla f(\mathbf{x}_t;z_{j_t})-\nabla F_S(\mathbf{x}_t)\| \sim \mathrm{subW}(\theta,K)$, we get $\mathbb{E}\left[\exp\left(\frac{\|\nabla f(\mathbf{x}_t;z_{j_t})-\nabla F_S(\mathbf{x}_t)\|^2}{K^2}\right)^{\frac{1}{2\theta}}\right]\leq 2$.

According to Lemma C.2, we get the following inequality with probability at least $1-\delta$

$$\sum_{t=1}^{T-1}\|\nabla f(\mathbf{x}_t;z_{j_t})-\nabla F_S(\mathbf{x}_t)\|^2 \leq (T-1)K^2g(2\theta)\log^{2\theta}(2/\delta).$$

Thus, with probability at least $1 - \delta$, we have

$$\sum_{t=1}^{T-1}(t + t_0 + 1)(t + t_0)\|\mathbf{m}_t\|^2 \leq \frac{(8C_\gamma)}{(1 - \gamma)\mu(S)^2}(T - 1)K^2g(2\theta)\log^{2\theta}(2/\delta)$$

$$+ \sum_{t=1}^{T-1}\frac{(8C_\gamma)}{(1 - \gamma)\mu(S)^2}\|\nabla F_S(\mathbf{x}_t)\|^2. \tag{31}$$

Similarly, with probability at least $1 - \delta$, we can derive

$$\sum_{t=1}^{T}(t + t_0)(t + t_0 - 1)\|\mathbf{m}_t\|^2$$

$$\leq \frac{(8C_\gamma)}{(1 - \gamma)\mu(S)^2}TK^2g(2\theta)\log^{2\theta}(2/\delta) + \sum_{t=1}^{T}\frac{(8C_\gamma)}{(1 - \gamma)\mu(S)^2}\|\nabla F_S(\mathbf{x}_t)\|^2.$$

We then bound $-\sum_{t=1}^{T}(t + t_0)(t + t_0 - 1)\langle\mathbf{m}_{t-1}, \nabla F_S(\mathbf{x}_{t-1})\rangle$. Recall that from (7), we know

$$-\langle\mathbf{m}_t, \nabla F_S(\mathbf{x}_t)\rangle \leq L\sum_{i=1}^{t-1}\gamma^{t-i}\|\mathbf{m}_i\|^2 - \sum_{i=1}^{t}\gamma^{t-i}\langle\eta_i\nabla f(\mathbf{x}_i; z_{j_i}), \nabla F_S(\mathbf{x}_i)\rangle.$$

Since $\mathbf{m}_0 = 0$, we have

$$-\sum_{t=1}^{T}(t + t_0)(t + t_0 - 1)\langle\mathbf{m}_{t-1}, \nabla F_S(\mathbf{x}_{t-1})\rangle$$

$$= -\sum_{t=1}^{T-1}(t + t_0 + 1)(t + t_0)\langle\mathbf{m}_t, \nabla F_S(\mathbf{x}_t)\rangle$$

$$\leq \sum_{t=1}^{T-1}(t + t_0 + 1)(t + t_0)L\sum_{i=1}^{t-1}\gamma^{t-i}\|\mathbf{m}_i\|^2$$

$$- \sum_{t=1}^{T-1}(t + t_0 + 1)(t + t_0)\sum_{i=1}^{t}\gamma^{t-i}\langle\eta_i\nabla f(\mathbf{x}_i; z_{j_i}), \nabla F_S(\mathbf{x}_i)\rangle$$

$$\leq \sum_{t=1}^{T-1}(t + t_0 + 1)(t + t_0)L\sum_{i=1}^{t}\gamma^{t-i}\|\mathbf{m}_i\|^2$$

$$- \sum_{t=1}^{T-1}(t + t_0 + 1)(t + t_0)\sum_{i=1}^{t}\gamma^{t-i}\langle\eta_i\nabla f(\mathbf{x}_i; z_{j_i}), \nabla F_S(\mathbf{x}_i)\rangle$$

$$= \sum_{t=1}^{T-1}\gamma^{-t}\|\mathbf{m}_t\|^2L\sum_{i=t}^{T-1}\gamma^i(i + t_0 + 1)(i + t_0)$$

$$- \sum_{t=1}^{T-1}\gamma^{-t}\langle\eta_t\nabla f(\mathbf{x}_t; z_{j_t}), \nabla F_S(\mathbf{x}_t)\rangle\sum_{i=t}^{T-1}(i + t_0 + 1)(i + t_0)\gamma^i$$

$$= \sum_{t=1}^{T-1}\gamma^{-t}\|\mathbf{m}_t\|^2L\sum_{i=t}^{T-1}\gamma^i(i + t_0 + 1)(i + t_0)$$

$$- \sum_{t=1}^{T-1}\gamma^{-t}\langle\eta_t(\nabla f(\mathbf{x}_t; z_{j_t}) - \nabla F_S(\mathbf{x}_t)), \nabla F_S(\mathbf{x}_t)\rangle\sum_{i=t}^{T-1}(i + t_0 + 1)(i + t_0)\gamma^i$$

$$- \sum_{t=1}^{T-1}\gamma^{-t}\langle\eta_t\nabla F_S(\mathbf{x}_t), \nabla F_S(\mathbf{x}_t)\rangle\sum_{i=t}^{T-1}(i + t_0 + 1)(i + t_0)\gamma^i,$$

where the second equation holds by using Lemma C.6.

With a similar analysis to (31), it is clear that with probability $1 - \delta$

$$\sum_{t=1}^{T-1} \gamma^{-t} \|\mathbf{m}_t\|^2 L \sum_{i=t}^{T-1} \gamma^i (i + t_0 + 1)(i + t_0) \leq L C_\gamma \sum_{t=1}^{T-1} \|\mathbf{m}_t\|^2 (t + t_0 + 1)^2$$

$$\leq L(C_\gamma) \frac{(8C_\gamma)}{(1 - \gamma)\mu(S)^2} (T - 1) K^2 g(2\theta) \log^{2\theta}(2/\delta) + \sum_{t=1}^{T-1} L(C_\gamma) \frac{(8C_\gamma)}{(1 - \gamma)\mu(S)^2} \|\nabla F_S(\mathbf{x}_t)\|^2.$$

And we also have

$$- \sum_{t=1}^{T-1} \gamma^{-t} \langle \eta_t \nabla F_S(\mathbf{x}_t), \nabla F_S(\mathbf{x}_t) \rangle \sum_{i=t}^{T-1} (i + t_0 + 1)(i + t_0)\gamma^i$$

$$\leq - \sum_{t=1}^{T-1} \gamma^{-t}(t + t_0 + 1)(t + t_0)\langle \eta_t \nabla F_S(\mathbf{x}_t), \nabla F_S(\mathbf{x}_t) \rangle \sum_{i=t}^{T-1} \gamma^i$$

$$\leq - \sum_{t=1}^{T-1} (t + t_0 + 1)(t + t_0)\langle \eta_t \nabla F_S(\mathbf{x}_t), \nabla F_S(\mathbf{x}_t) \rangle$$

$$= - \sum_{t=1}^{T-1} (t + t_0 + 1)(t + t_0)\eta_t \|\nabla F_S(\mathbf{x}_t)\|^2.$$

Thus, we have

$$- \sum_{t=1}^{T} (t + t_0)(t + t_0 - 1)\langle \mathbf{m}_{t-1}, \nabla F_S(\mathbf{x}_{t-1}) \rangle$$

$$\leq - \sum_{t=1}^{T-1} \gamma^{-t} \langle \eta_t(\nabla f(\mathbf{x}_t; z_{j_t}) - \nabla F_S(\mathbf{x}_t)), \nabla F_S(\mathbf{x}_t) \rangle \sum_{i=t}^{T-1} (i + t_0 + 1)(i + t_0)\gamma^i$$

$$- \sum_{t=1}^{T-1} (t + t_0 + 1)(t + t_0)\eta_t \|\nabla F_S(\mathbf{x}_t)\|^2 + L(C_\gamma) \frac{(8C_\gamma)}{(1 - \gamma)\mu(S)^2} (T - 1) K^2 g(2\theta) \log^{2\theta}(2/\delta)$$

$$+ \sum_{t=1}^{T-1} L(C_\gamma) \frac{(8C_\gamma)}{(1 - \gamma)\mu(S)^2} \|\nabla F_S(\mathbf{x}_t)\|^2.$$

We now consider the term $- \sum_{t=1}^{T-1} \gamma^{-t} \langle \eta_t(\nabla f(\mathbf{x}_t; z_{j_t}) - \nabla F_S(\mathbf{x}_t)), \nabla F_S(\mathbf{x}_t) \rangle \sum_{i=t}^{T-1} (i + t_0 + 1)(i + t_0)\gamma^i$. Denoted by $\xi_t = -\gamma^{-t} \langle \eta_t(\nabla f(\mathbf{x}_t; z_{j_t}) - \nabla F_S(\mathbf{x}_t)), \nabla F_S(\mathbf{x}_t) \rangle \sum_{i=t}^{T-1} (i + t_0 + 1)(i + t_0)\gamma^i$. We know that $\mathbb{E}_{j_t} \xi_t = -\mathbb{E}_{j_t} \gamma^{-t} \langle \eta_t(\nabla f(\mathbf{x}_t; z_{j_t}) - \nabla F_S(\mathbf{x}_t)), \nabla F_S(\mathbf{x}_t) \rangle \sum_{i=t}^{T-1} (i + t_0 + 1)(i + t_0)\gamma^i = 0$, implying that it is a martingale difference sequence. We use Lemma C.4 to bound this term.

From (30), it is clear that $|\gamma^{-t} \langle \eta_t(\nabla f(\mathbf{x}_t; z_{j_t}) - \nabla F_S(\mathbf{x}_t)), \nabla F_S(\mathbf{x}_t) \rangle \sum_{i=t}^{T-1} (i + t_0 + 1)(i + t_0)\gamma^i| \leq (C_\gamma)(t + t_0 + 1)^2 \eta_t \|\nabla f(\mathbf{x}_t; z_{j_t}) - \nabla F_S(\mathbf{x}_t)\| \|\nabla F_S(\mathbf{x}_t)\|$. We set

$$K_{t-1} = C_\gamma(t + t_0 + 1)^2 \eta_t K \|\nabla F_S(\mathbf{x}_t)\| = C_\gamma(t + t_0 + 1)^2 \frac{1}{\mu(S)(t + t_0)} K \|\nabla F_S(\mathbf{x}_t)\|.$$

We also set $\beta = 0$, $\lambda = \frac{1}{2\alpha}$, and $x = 2\alpha \log(1/\delta)$. For brevity, we denote $\Xi = 2C_\gamma(t + t_0 + 1)\mu(S)^{-1}K$ and $\Xi_T = 2C_\gamma(T + t_0 + 1)\mu(S)^{-1}K$. Moreover, according to the smoothness assumption, we know $\|\nabla F_S(\mathbf{x}_t)\| \leq (L\|\mathbf{x}_t\| + B)$.

If $\theta = \frac{1}{2}$, for all $\alpha > 0$, we have the following inequality with probability $1 - \delta$

$$- \sum_{t=1}^{T-1} \gamma^{-t} \langle \eta_t \nabla f(\mathbf{x}_t; z_{j_t}), \nabla F_S(\mathbf{x}_t) \rangle \sum_{i=t}^{T-1} (i + t_0 + 1)(i + t_0)\gamma^i$$

$$\leq 2\alpha \log(1/\delta) + \frac{a}{\alpha} \sum_{t=1}^{T-1} \Xi^2 \|\nabla F_S(\mathbf{x}_t)\|^2.$$

If $\frac{1}{2} < \theta \le 1$, we set $m_t = \Xi(L\|\mathbf{x}_t\| + B)$. Then for all $\alpha \ge b\Xi_T(L\|\mathbf{x}_T\| + B)$, we have the following inequality with probability $1 - \delta$

$$
-\sum_{t=1}^{T-1} \gamma^{-t}\langle \eta_t \nabla f(\mathbf{x}_t; z_{j_t}), \nabla F_S(\mathbf{x}_t)\rangle \sum_{i=t}^{T-1}(i + t_0 + 1)(i + t_0)\gamma^i
$$

$$
\le 2\alpha \log(1/\delta) + \frac{a}{\alpha}\sum_{t=1}^{T-1}\Xi^2\|\nabla F_S(\mathbf{x}_t)\|^2.
$$

If $\theta > 1$, we set $m_t = \Xi(L\|\mathbf{x}_t\| + B)$ and $\delta = \delta$. Then, for all $\alpha \ge b\Xi_T(L\|\mathbf{x}_T\| + B)$, we have the following inequality with probability $1 - 3\delta$

$$
-\sum_{t=1}^{T-1} \gamma^{-t}\langle \eta_t \nabla f(\mathbf{x}_t; z_{j_t}), \nabla F_S(\mathbf{x}_t)\rangle \sum_{i=t}^{T-1}(i + t_0 + 1)(i + t_0)\gamma^i
$$

$$
\le 2\alpha \log(1/\delta) + \frac{a}{\alpha}\sum_{t=1}^{T-1}\Xi^2\|\nabla F_S(\mathbf{x}_t)\|^2.
$$

We now consider the last term $-(t + t_0)(t + t_0 - 1)\eta_t\langle \nabla f(\mathbf{x}_t; z_{j_t}) - \nabla F_S(\mathbf{x}_t), \nabla F_S(\mathbf{x}_t)\rangle$. With a similar analysis, we set $\xi_t = -(t + t_0)(t + t_0 - 1)\eta_t\langle \nabla f(\mathbf{x}_t; z_{j_t}) - \nabla F_S(\mathbf{x}_t), \nabla F_S(\mathbf{x}_t)\rangle$ and

$$
K_{t-1} = (t + t_0)(t + t_0 - 1)\eta_t K\|\nabla F_S(\mathbf{x}_t)\| = \mu(S)^{-1}(t + t_0 - 1)K\|\nabla F_S(\mathbf{x}_t)\|.
$$

We also set $\beta = 0$, $\lambda = \frac{1}{2\alpha}$, and $x = 2\alpha \log(1/\delta)$. According to the smoothness assumption, we know $\|\nabla F_S(\mathbf{x}_t)\| \le (L\|\mathbf{x}_t\| + B)$.

If $\theta = \frac{1}{2}$, for all $\alpha > 0$, we have the following inequality with probability at least $1 - \delta$

$$
-\sum_{t=1}^{T}(t + t_0)(t + t_0 - 1)\eta_t\langle \nabla f(\mathbf{x}_t; z_{j_t}) - \nabla F_S(\mathbf{x}_t), \nabla F_S(\mathbf{x}_t)\rangle
$$

$$
\le 2\alpha \log(1/\delta) + \frac{aK^2}{\mu(S)^2\alpha}\sum_{t=1}^{T}(t + t_0 - 1)^2\|\nabla F_S(\mathbf{x}_t)\|^2.
$$

If $\frac{1}{2} < \theta \le 1$, we set $m_t = \mu(S)^{-1}(t + t_0 - 1)K(L\|\mathbf{x}_t\| + B)$. Then for all $\alpha \ge b\mu(S)^{-1}(T + t_0 - 1)K(L\|\mathbf{x}_T\| + B)$, we have the following inequality with probability at least $1 - \delta$

$$
-\sum_{t=1}^{T}(t + t_0)(t + t_0 - 1)\eta_t\langle \nabla f(\mathbf{x}_t; z_{j_t}) - \nabla F_S(\mathbf{x}_t), \nabla F_S(\mathbf{x}_t)\rangle
$$

$$
\le 2\alpha \log(1/\delta) + \frac{aK^2}{\mu(S)^2\alpha}\sum_{t=1}^{T}(t + t_0 - 1)^2\|\nabla F_S(\mathbf{x}_t)\|^2.
$$

If $\theta > 1$, we set $m_t = \mu(S)^{-1}(t + t_0 - 1)K(L\|\mathbf{x}_t\| + B)$ and $\delta = \delta$. Then, for all $\alpha \ge b\mu(S)^{-1}(T + t_0 - 1)K(L\|\mathbf{x}_T\| + B)$, we have the following inequality with probability at least $1 - 3\delta$

$$
-\sum_{t=1}^{T}(t + t_0)(t + t_0 - 1)\eta_t\langle \nabla f(\mathbf{x}_t; z_{j_t}) - \nabla F_S(\mathbf{x}_t), \nabla F_S(\mathbf{x}_t)\rangle
$$

$$
\le 2\alpha \log(1/\delta) + \frac{aK^2}{\mu(S)^2\alpha}\sum_{t=1}^{T}(t + t_0 - 1)^2\|\nabla F_S(\mathbf{x}_t)\|^2.
$$

Finally, combining with these terms, we derive

$$\sum_{t=1}^{T} \frac{(t+t_0-1)}{2\mu(S)} \|\nabla F_S(\mathbf{x}_t)\|^2 - \frac{aK^2}{\mu(S)^2\alpha} \sum_{t=1}^{T} (t+t_0-1)^2 \|\nabla F_S(\mathbf{x}_t)\|^2$$

$$- \frac{L}{2} \sum_{t=1}^{T} \frac{(8C_\gamma)}{(1-\gamma)\mu(S)^2} \|\nabla F_S(\mathbf{x}_t)\|^2$$

$$- L\gamma \sum_{t=1}^{T-1} \frac{(8C_\gamma)}{(1-\gamma)\mu(S)^2} \|\nabla F_S(\mathbf{x}_t)\|^2 + \sum_{t=1}^{T-1} (t+t_0+1)(t+t_0)\eta_t \|\nabla F_S(\mathbf{x}_t)\|^2$$

$$- \sum_{t=1}^{T-1} L\gamma(C_\gamma) \frac{(8C_\gamma)}{(1-\gamma)\mu(S)^2} \|\nabla F_S(\mathbf{x}_t)\|^2$$

$$- \gamma \frac{a}{\alpha} \sum_{t=1}^{T-1} \Xi^2 \|\nabla F_S(\mathbf{x}_t)\|^2 + (T+t_0)(T+t_0-1)(F_S(\mathbf{x}_{T+1}) - F_S(\mathbf{x}(S)))$$

$$\leq L\gamma \frac{(8C_\gamma)}{(1-\gamma)\mu(S)^2} (T-1)K^2 g(2\theta) \log^{2\theta}(2/\delta) + \frac{L}{2} \frac{(8C_\gamma)}{(1-\gamma)\mu(S)^2} TK^2 g(2\theta) \log^{2\theta}(2/\delta)$$

$$+ L\gamma(C_\gamma) \frac{(8C_\gamma)}{(1-\gamma)\mu(S)^2} (T-1)K^2 g(2\theta) \log^{2\theta}(2/\delta) + (t_0-1)(t_0-2)(F_S(\mathbf{x}_1) - F_S(\mathbf{x}(S)))$$

$$+ 2\alpha \log(1/\delta) + \gamma 2\alpha \log(1/\delta). \tag{32}$$

We want

$$\frac{(t+t_0-1)}{2\mu(S)} - \frac{aK^2}{\mu(S)^2\alpha}(t+t_0-1)^2 - \frac{L}{2} \frac{(8C_\gamma)}{(1-\gamma)^2\mu(S)^2} \geq 0$$

and

$$\frac{(t+t_0+1)}{\mu(S)} - L\gamma \frac{(8C_\gamma)}{(1-\gamma)\mu(S)^2} - L\gamma(C_\gamma) \frac{(8C_\gamma)}{(1-\gamma)\mu(S)^2} - \gamma \frac{a}{\alpha} \Xi^2 \geq 0.$$

Thus, we assume that $t_0$ satisfies the following conditions

$$\frac{(t_0-1)}{2\mu(S)} \geq \frac{L}{2} \frac{(8C_\gamma)}{(1-\gamma)^2\mu(S)^2};$$

and

$$\frac{(t_0+1)}{\mu(S)} \geq L\gamma \frac{(8C_\gamma)}{(1-\gamma)\mu(S)^2} + L\gamma(C_\gamma) \frac{(8C_\gamma)}{(1-\gamma)\mu(S)^2},$$

which means that

$$t_0 \geq \frac{(8C_\gamma)L}{(1-\gamma)^2\mu(S)} + 1;$$

and

$$t_0 \geq \frac{8C_\gamma(L\gamma + L\gamma(C_\gamma))}{(1-\gamma)\mu(S)} - 1.$$

Thus, we can further derive that $\alpha \geq \frac{aK^2(t+t_0-1)^2}{\frac{(t+t_0-1)}{2\mu(S)} - \frac{L}{2}\frac{(8C_\gamma)}{(1-\gamma)^2\mu(S)^2}}$ and

$$\alpha \geq \frac{\gamma a(2C_\gamma(t+t_0+1)\mu(S)^{-1}K)^2}{\frac{(t+t_0+1)}{\mu(S)} - L\gamma\frac{(8C_\gamma)}{(1-\gamma)^2\mu(S)^2} - L\gamma(C_\gamma)\frac{(8C_\gamma)}{(1-\gamma)^2\mu(S)^2}}.$$

When $\theta = \frac{1}{2}$, the above lower bounds of $\alpha$ are: $\alpha \geq \frac{aK^2(t+t_0-1)^2}{\frac{(t+t_0-1)}{2\mu(S)} - \frac{L}{2}\frac{(8C_\gamma)}{(1-\gamma)^2\mu(S)^2}}$,

$\alpha \geq \frac{\gamma a(2C_\gamma(t+t_0+1)\mu(S)^{-1}K)^2}{\frac{(t+t_0+1)}{\mu(S)} - L\gamma\frac{(8C_\gamma)}{(1-\gamma)^2\mu(S)^2} - L\gamma(C_\gamma)\frac{(8C_\gamma)}{(1-\gamma)^2\mu(S)^2}}$, and $\alpha > 0$, which implies that we should choose $\alpha = \mathcal{O}(T)$.

When $\frac{1}{2} < \theta \leq 1$, the above lower bounds of $\alpha$ are: $\alpha \geq \frac{aK^2(t+t_0-1)^2}{\frac{(t+t_0-1)}{2\mu(S)} - \frac{L}{2}\frac{(8C_\gamma)}{(1-\gamma)^2\mu(S)^2}}$, $\alpha \geq$
$\frac{\gamma a(2C_\gamma(t+t_0+1)\mu(S)^{-1}K)^2}{\frac{(t+t_0+1)}{\mu(S)} - L\gamma\frac{(8C_\gamma)}{(1-\gamma)^2\mu(S)^2} - L\gamma(C_\gamma)\frac{(8C_\gamma)}{(1-\gamma)^2\mu(S)^2}}$, $\alpha \geq b\Xi_T(L\|\mathbf{x}_T\| + B)$, and $\alpha \geq b\mu(S)^{-1}(T + t_0 - 1)K(L\|\mathbf{x}_T\| + B)$, which implies that we should choose $\alpha = \mathcal{O}\left(T\log^{(\theta+\frac{1}{2})}(\frac{1}{\delta})\log^{\frac{1}{2}}T\right)$.

When $\theta > 1$, the above lower bounds of $\alpha$ are: $\alpha \geq \frac{aK^2(t+t_0-1)^2}{\frac{(t+t_0-1)}{2\mu(S)} - \frac{L}{2}\frac{(8C_\gamma)}{(1-\gamma)^2\mu(S)^2}}$,

$\alpha \geq \frac{\gamma a(2C_\gamma(t+t_0+1)\mu(S)^{-1}K)^2}{\frac{(t+t_0+1)}{\mu(S)} - L\gamma\frac{(8C_\gamma)}{(1-\gamma)^2\mu(S)^2} - L\gamma(C_\gamma)\frac{(8C_\gamma)}{(1-\gamma)^2\mu(S)^2}}$, $\alpha \geq b\Xi_T(L\|\mathbf{x}_T\| + B)$, and $\alpha \geq b\mu(S)^{-1}(T + t_0 - 1)K(L\|\mathbf{x}_T\| + B)$, which implies that we should choose

$$\alpha = \mathcal{O}\left(\log^{\theta-1}(\frac{T}{\delta})T\left(\log^{(\theta+\frac{1}{2})}(\frac{1}{\delta}) + \log^{\frac{\theta-1}{2}}(T/\delta)\log^{\frac{1}{2}}(1/\delta)\right)\log^{\frac{1}{2}}T\right).$$

Note that the bound of $\|\mathbf{x}_T\|$ comes from (27).

Thus, we derive that

$$(T + t_0)(T + t_0 - 1)(F_S(\mathbf{x}_{t+1}) - F_S(\mathbf{x}(S)))$$
$$\leq L\gamma\frac{(8C_\gamma)}{(1-\gamma)\mu(S)^2}(T-1)K^2 g(2\theta)\log^{2\theta}(2/\delta) + \frac{L}{2}\frac{(8C_\gamma)}{(1-\gamma)\,\mu(S)^2}TK^2 g(2\theta)\log^{2\theta}(2/\delta)$$
$$+ L\gamma(C_\gamma)\frac{(8C_\gamma)}{(1-\gamma)\mu(S)^2}(T-1)K^2 g(2\theta)\log^{2\theta}(2/\delta)$$
$$+ (t_0 - 1)(t_0 - 2)(F_S(\mathbf{x}_1) - F_S(\mathbf{x}(S))) + 2\alpha\log(1/\delta) + \gamma 2\alpha\log(1/\delta).$$

Putting the previous bounds together.

If $\theta = 1$, with probability $1 - 6\delta$, we have

$$F_S(\mathbf{x}_{T+1}) - F_S(\mathbf{x}(S)) = \mathcal{O}\left(\frac{\log(1/\delta)}{T}\right).$$

If $\frac{1}{2} < \theta \leq 1$, with probability $1 - 7\delta$, we have

$$F_S(\mathbf{x}_{T+1}) - F_S(\mathbf{x}(S)) = \mathcal{O}\left(\frac{\log^{(\theta+\frac{1}{2})}(\frac{1}{\delta})\log^{\frac{1}{2}}T}{T}\log(\frac{1}{\delta})\right).$$

If $\theta > 1$, with probability $1 - 10\delta$, we have

$$F_S(\mathbf{x}_{T+1}) - F_S(\mathbf{x}(S)) = \mathcal{O}\left(\frac{\left(\log^{(\theta+\frac{1}{2})}(\frac{1}{\delta}) + \Delta^{\frac{1}{2}}(\theta, T, \delta)\right)\log^{\frac{1}{2}}T}{T}\log^{\theta-1}(\frac{T}{\delta})\log(\frac{1}{\delta})\right).$$

The above bounds mean that with probability $1 - \delta$, there holds

$$F_S(\mathbf{x}_{T+1}) - F_S(\mathbf{x}(S)) = \begin{cases} \mathcal{O}\left(\frac{\log(1/\delta)}{T}\right) & \text{if} \quad \theta = \frac{1}{2}, \\ \mathcal{O}\left(\frac{\log^{(\theta+\frac{3}{2})}(\frac{1}{\delta})\log^{\frac{1}{2}}T}{T}\right) & \text{if} \quad \theta \in (\frac{1}{2}, 1], \\ \mathcal{O}\left(\frac{\log^{(\theta+\frac{3}{2})}(\frac{1}{\delta})\log^{\frac{3(\theta-1)}{2}}(T/\delta)\log^{\frac{1}{2}}T}{T}\right) & \text{if} \quad \theta > 1. \end{cases} \tag{33}$$

The proof is complete. $\qquad\square$

### D.4 Proof of Theorem 3.7

*Proof.* Recall Assumption 2.7 (Polyak-Łojasiewicz condition), which gives

$$F(\mathbf{x}_{T+1}) - F(\mathbf{x}^*) \leq \frac{1}{4\mu}\|\nabla F(\mathbf{x}_{T+1})\|^2 \leq \frac{1}{2\mu}(\|\nabla F(\mathbf{x}_{T+1}) - \nabla F_S(\mathbf{x}_{T+1})\|^2 + \|\nabla F_S(\mathbf{x}_{T+1})\|^2).$$
$$\tag{34}$$

From (27) and Lemma C.8, with probability $1 - \delta$ we have

$$\|\nabla F(\mathbf{x}_{T+1}) - \nabla F_S(\mathbf{x}_{T+1})\|^2 = \mathcal{O}\left(\frac{d + \log(\frac{1}{\delta})}{n}\|\mathbf{x}_{T+1}\|^2\right)$$

$$= \mathcal{O}\left(\frac{d + \log(\frac{1}{\delta})}{n}\left(\log^{(2\theta+1)}(\frac{1}{\delta}) + \Delta(\theta, T, \delta)\right)\log T\right). \tag{35}$$

From the smoothness property in Lemma C.7 and the convergence bound in (33), with probability $1 - \delta$, there holds

$$\|\nabla F_S(\mathbf{x}_{T+1})\|^2 \leq (2L)(F_S(\mathbf{x}_{T+1}) - F_S(\mathbf{x}(S)))$$

$$= \begin{cases} \mathcal{O}\left(\frac{\log(1/\delta)}{T}\right) & \text{if} \quad \theta = \frac{1}{2}, \\ \mathcal{O}\left(\frac{\log^{(\theta+\frac{3}{2})}(\frac{1}{\delta})\log^{\frac{1}{2}}T}{T}\right) & \text{if} \quad \theta \in (\frac{1}{2}, 1], \\ \mathcal{O}\left(\frac{\log^{(\theta+\frac{3}{2})}(\frac{1}{\delta})\log^{\frac{3(\theta-1)}{2}}(T/\delta)\log^{\frac{1}{2}}T}{T}\right) & \text{if} \quad \theta > 1. \end{cases} \tag{36}$$

Plugging (35) and (36) into (34), we derive that with probability $1 - 2\delta$, there holds: (1.) if $\theta = \frac{1}{2}$,

$$F(\mathbf{x}_{T+1}) - F(\mathbf{x}^*) = \mathcal{O}\left(\frac{\log(1/\delta)}{T} + \frac{d + \log(\frac{1}{\delta})}{n}\log^2(\frac{1}{\delta})\log T\right);$$

(2.) if $\theta \in (\frac{1}{2}, 1]$,

$$F(\mathbf{x}_{T+1}) - F(\mathbf{x}^*) = \mathcal{O}\left(\frac{\log^{(\theta+\frac{3}{2})}(\frac{1}{\delta})\log^{\frac{1}{2}}T}{T} + \frac{d + \log(\frac{1}{\delta})}{n}\log^{(2\theta+1)}(\frac{1}{\delta})\log T\right);$$

(3.) if $\theta > 1$,

$$F(\mathbf{x}_{T+1}) - F(\mathbf{x}^*)$$

$$= \mathcal{O}\left(\frac{\log^{(\theta+\frac{3}{2})}(\frac{1}{\delta})\log^{\frac{3(\theta-1)}{2}}(T/\delta)\log^{\frac{1}{2}}T}{T} + \frac{d + \log(\frac{1}{\delta})}{n}\left(\log^{(2\theta+1)}(\frac{1}{\delta}) + \Delta(\theta, T, \delta)\right)\log T\right).$$

We choose $T \asymp n$, then with probability at least $1 - \delta$, there holds

$$F(\mathbf{x}_{T+1}) - F(\mathbf{x}^*) = \begin{cases} \mathcal{O}\left(\frac{d + \log(\frac{1}{\delta})}{n}\log^2(\frac{1}{\delta})\log n\right) & \text{if} \quad \theta = \frac{1}{2}, \\ \mathcal{O}\left(\frac{d + \log(\frac{1}{\delta})}{n}\log^{(2\theta+1)}(\frac{1}{\delta})\log n\right) & \text{if} \quad \theta \in (\frac{1}{2}, 1], \\ \mathcal{O}\left(\frac{d + \log(\frac{1}{\delta})}{n}\log^{(2\theta+1)}(\frac{1}{\delta})\log^{\frac{3(\theta-1)}{2}}(\frac{n}{\delta})\log n\right) & \text{if} \quad \theta > 1. \end{cases}$$

The proof is complete. $\qquad\qquad\square$

### D.5 PROOF OF THEOREM 3.9

*Proof.* By Lemma C.9, with probability $1 - \delta$ we have

$$\|\nabla F(\mathbf{w}_{T+1}) - \nabla F_S(\mathbf{w}_{T+1})\|^2$$

$$\leq \left(\|\nabla F_S(\mathbf{w}_{T+1})\| + \frac{\mu}{n} + 2\frac{B_* \log(4/\delta)}{n} + 2\sqrt{\frac{2\mathbb{E}[\|\nabla f(\mathbf{x}^*; z)\|^2]\log(4/\delta)}{n}}\right)^2$$

$$\leq 4\left(\|\nabla F_S(\mathbf{w}_{T+1})\|^2 + 4\frac{B_*^2 \log^2(4/\delta)}{n^2} + 8\frac{\mathbb{E}[\|\nabla f(\mathbf{x}^*; z)\|^2]\log(4/\delta)}{n} + \frac{\mu^2}{n^2}\right).$$

From the smoothness property in Lemma C.7, if $f$ is nonnegative and $L$-smooth, we have $\|\nabla f(\mathbf{x}^*; z)\|^2 \leq 2L\nabla f(\mathbf{x}^*; z)$, implying that $\mathbb{E}[\|\nabla f(\mathbf{x}^*; z)\|^2] \leq 2LF(\mathbf{x}^*)$. Thus, with probability $1 - \delta$ we have

$$\|\nabla F(\mathbf{w}_{T+1}) - \nabla F_S(\mathbf{w}_{T+1})\|^2$$

$$\leq 4\left(\|\nabla F_S(\mathbf{w}_{T+1})\|^2 + 4\frac{B_*^2 \log^2(4/\delta)}{n^2} + \frac{16LF(\mathbf{x}^*)\log(4/\delta)}{n} + \frac{\mu^2}{n^2}\right). \tag{37}$$

Again, from the smoothness property in Lemma C.7 and the convergence bound in (33), with probability $1 - \delta$, there holds

$$\|\nabla F_S(\mathbf{x}_{T+1})\|^2 \leq (2L)(F_S(\mathbf{x}_{T+1}) - F_S(\mathbf{x}(S)))$$

$$= \begin{cases} \mathcal{O}\left(\frac{\log(1/\delta)}{T}\right) & \text{if} \quad \theta = \frac{1}{2}, \\ \mathcal{O}\left(\frac{\log^{(\theta+\frac{3}{2})}(\frac{1}{\delta})\log^{\frac{1}{2}}T}{T}\right) & \text{if} \quad \theta \in (\frac{1}{2}, 1], \\ \mathcal{O}\left(\frac{\log^{(\theta+\frac{3}{2})}(\frac{1}{\delta})\log^{\frac{3(\theta-1)}{2}}(T/\delta)\log^{\frac{1}{2}}T}{T}\right) & \text{if} \quad \theta > 1. \end{cases} \tag{38}$$

Plugging (38) into (37), with probability $1 - 2\delta$, we have: (1.) if $\theta = \frac{1}{2}$,

$$\|\nabla F(\mathbf{w}_{T+1}) - \nabla F_S(\mathbf{w}_{T+1})\|^2 = \mathcal{O}\left(\frac{\log(1/\delta)}{T} + \frac{\log^2(1/\delta)}{n^2} + \frac{F(\mathbf{x}^*)\log(1/\delta)}{n}\right); \tag{39}$$

(2.) if $\theta \in (\frac{1}{2}, 1]$,

$$\|\nabla F(\mathbf{w}_{T+1}) - \nabla F_S(\mathbf{w}_{T+1})\|^2 = \mathcal{O}\left(\frac{\log^{(\theta+\frac{3}{2})}(\frac{1}{\delta})\log^{\frac{1}{2}}T}{T} + \frac{\log^2(1/\delta)}{n^2} + \frac{F(\mathbf{x}^*)\log(1/\delta)}{n}\right); \tag{40}$$

(3.) if $\theta > 1$,

$$\|\nabla F(\mathbf{w}_{T+1}) - \nabla F_S(\mathbf{w}_{T+1})\|^2$$
$$= \mathcal{O}\left(\frac{\log^{(\theta+\frac{3}{2})}(\frac{1}{\delta})\log^{\frac{3(\theta-1)}{2}}(T/\delta)\log^{\frac{1}{2}}T}{T} + \frac{\log^2(1/\delta)}{n^2} + \frac{F(\mathbf{x}^*)\log(1/\delta)}{n}\right). \tag{41}$$

According to the Polyak-Łojasiewicz condition, we know

$$F(\mathbf{w}_{T+1}) - F(\mathbf{x}^*) \leq \frac{1}{4\mu}\|\nabla F(\mathbf{w}_{T+1})\|^2$$
$$\leq (2\mu)^{-1}(\|\nabla F(\mathbf{w}_{T+1}) - \nabla F_S(\mathbf{w}_{T+1})\|^2 + \|\nabla F_S(\mathbf{w}_{T+1})\|^2). \tag{42}$$

Plugging the convergence bound in (38) and the generalization bound in (39)-(41) into (42), with probability $1 - 3\delta$, we have (1.) if $\theta = \frac{1}{2}$,

$$F(\mathbf{w}_{T+1}) - F(\mathbf{x}^*) = \mathcal{O}\left(\frac{\log(1/\delta)}{T} + \frac{\log^2(1/\delta)}{n^2} + \frac{F(\mathbf{x}^*)\log(1/\delta)}{n}\right);$$

(2.) if $\theta \in (\frac{1}{2}, 1]$,

$$F(\mathbf{w}_{T+1}) - F(\mathbf{x}^*) = \mathcal{O}\left(\frac{\log^{(\theta+\frac{3}{2})}(\frac{1}{\delta})\log^{\frac{1}{2}}T}{T} + \frac{\log^2(1/\delta)}{n^2} + \frac{F(\mathbf{x}^*)\log(1/\delta)}{n}\right);$$

(3.) if $\theta > 1$,

$$F(\mathbf{w}_{T+1}) - F(\mathbf{x}^*) = \mathcal{O}\left(\frac{\log^{(\theta+\frac{3}{2})}(\frac{1}{\delta})\log^{\frac{3(\theta-1)}{2}}(T/\delta)\log^{\frac{1}{2}}T}{T} + \frac{\log^2(1/\delta)}{n^2} + \frac{F(\mathbf{x}^*)\log(1/\delta)}{n}\right).$$

We choose $T \asymp n^2$, then the following inequality holds with probability $1 - \delta$

$$F(\mathbf{w}_{T+1}) - F(\mathbf{x}^*) = \begin{cases} \mathcal{O}\left(\frac{\log^2(1/\delta)}{n^2} + \frac{F(\mathbf{x}^*)\log(1/\delta)}{n}\right) & \text{if } \theta = \frac{1}{2}, \\ \mathcal{O}\left(\frac{\log^{(\theta+\frac{3}{2})}(\frac{1}{\delta})\log^{\frac{1}{2}}n}{n^2} + \frac{F(\mathbf{x}^*)\log(1/\delta)}{n}\right) & \text{if } \theta \in (\frac{1}{2}, 1], \\ \mathcal{O}\left(\frac{\log^{\frac{3(\theta-1)}{2}}(n/\delta)\log^{(\theta+\frac{3}{2})}(\frac{1}{\delta})\log^{\frac{1}{2}}n}{n^2} + \frac{F(\mathbf{x}^*)\log(1/\delta)}{n}\right) & \text{if } \theta > 1. \end{cases}$$

The proof is complete. □

