# OpenReview forum: "High Probability Bounds for Non-Convex Stochastic Optimization with Momentum"
_ICLR.cc/2026/Conference — ICLR 2026 Poster_

### Official Review · Reviewer_jh4H · 2025-10-30

**Soundness:** 2
**Presentation:** 3
**Contribution:** 2
**Rating:** 4
**Confidence:** 3

**Summary:**

This paper establishes high probability convergence and generalization upper bounds for non-convex SGDM. Specifically, it first considers the general non-convex case. The convergence results are better than previous works and the generalization bounds are the first ones for SGDM. Then, when PL condition holds, faster rates are achieved than the general non-convex case. Finally, further assuming Bernstein condition brings sharper generalization bounds in the case of low noise.

**Strengths:**

1.This paper utilizes some special properties of a heavy-tailed distribution, Sub-Weibull distribution, to develop the theoretical analysis of SGDM.

2.The whole paper is written clearly and easy to be understood.

**Weaknesses:**

1.The inequality in line 1240 may not be true, considering that $(1-\gamma^{t-i+1})$ is the coefficient of the gradient $\nabla F_S(x_i)$. We can not ensure the norm with $(1-\gamma^t)$ is larger than the one with $(1-\gamma^{t-i+1})$. Therefore, it may greatly affect the subsequent proof.

2.It is better to make a table to list all convergence and generalization results and all previous results, which facilitates readers' understanding of the advantages of their results over previous works.

**Questions:**

1.Why do the authors use two symbols $f$ and $g$ in Assumption 2.1?

2.There are some typos, such as “smothness”in the last line of page 3 and $k=2,...，$ in Remark 2.3.

3.The whole analysis framework is very similar to [1] except for the lemma which interchanges the order of summation (Lemma C.6) to enable the analysis of [1] in the SGDM case. Due to this reason, all results are the same as those of [1]. Is my understanding right? If so, do these results give us any insights?

Note: From my perspective, the major innovation of proof is mainly presented in Theorem 3.1 and Theorem 3.3, which is listed in **Q3**. If any other innovations exist in the proofs of other results compared with [1], please make them clear. If so, I will improve my score.

---

> ### Author Response · Authors · 2025-11-20
>
> **Dear Reviewer jh4H**, we sincerely appreciate your invaluable and constructive comments, which are very helpful to improve the paper. In light of the reviewers’ comments, we have thoroughly revised the manuscript, carefully going through it almost line by line to improve its clarity, readability, and overall presentation; major revisions are highlighted in blue for ease of reference.
>
>
> **R1: The inequality in line 1240 may not be true, considering that $(1-\gamma^{t-i+1})$ is the coefficient of the gradient $\nabla F_S(x_i)$. We cannot ensure the norm with $(1-\gamma^t)$ is larger than the one with $(1-\gamma^{t-i+1})$. Therefore, it may greatly affect the subsequent proof.**
>
>
> **A.** We appreciate the reviewer’s careful check. You are absolutely right that the
> inequality in the original line 1240, which implicitly replaced the coefficients
> $(1-\gamma^{t-i+1})$ by $(1-\gamma^t)$, is not valid. In the revised
> manuscript we have removed this incorrect step and replaced it with a different,
> correct bound.
>
> More precisely, in the new version we keep the exact coefficients and write
> (cf.\ Eq. (18)):
> \begin{align*}
> \\|x_{t+1}\\|\le \frac{1}{1-\gamma}\Big\\|\sum_{i=1}^t (1-\gamma^{t-i+1})\eta_i
>   \big(\nabla f(x_i;z_{j_i})-\nabla F_S(x_i)\big)\Big\\|  + \frac{1}{1-\gamma}\Big\\|\sum_{i=1}^t (1-\gamma^{t-i+1})\eta_i
>   \nabla F_S(x_i)\Big\\|.
> \end{align*}
> For the second term we now use a weighted Cauchy--Schwarz inequality. Let
> $a_i=(1-\gamma^{t-i+1})\eta_i $ and $v_i=\nabla F_S(x_i)$, then
> \begin{align*}
>   \Big\\|\sum_{i=1}^t a_i v_i\Big\\|^2
>   \le \Big(\sum_{i=1}^t a_i\Big)\Big(\sum_{i=1}^t a_i\\|v_i\\|^2\Big).
> \end{align*}
> Since $0 < 1-\gamma^{t-i+1} < 1$ and $\eta_i > 0$, this yields (Eq. (21)):
> \begin{align*}
>   \Big\\|\sum_{i=1}^t (1-\gamma^{t-i+1})\eta_i\nabla F_S(x_i)\Big\\|^2
>   \le
>   \Big(\sum_{i=1}^t\eta_i\Big)\Big(\sum_{i=1}^t\eta_i\\|\nabla F_S(x_i)\\|^2\Big).
> \end{align*}
> This is the bound we need in the original submission, thus the mistake in the original line 1240 does not affect any of our stated results. We have revised the proof and the surrounding text accordingly
> in the new version.
>
> **R2: It is better to make a table to list all convergence and generalization results and all previous results, which facilitates readers' understanding of the advantages of their results over previous works.**
>
>
> **A.** We thank the reviewer for this helpful suggestion. The submission contained a summary table (Table 1) and a separate section (Appendix B; summary of results), where we listed our bounds and  the most relevant results of SGDM. In the revised version, we explicitly point readers to Table 1 and Appendix B in the organization paragraph of the Introduction section.
>
>
> **Q1: Why do the authors use two symbols $f$ and $g$ in Assumption 2.1?**
>
>
> **A.** We agree that this notation is unnecessarily confusing. There is no structural need to distinguish $f$ and $g$ in Assumption 2.1; a single symbol is sufficient. In the revision we have rewrote Assumption 2.1 using a single function symbol throughout.
>
> **Q2: There are some typos, such as `smothness' in the last line of page 3 and $k=2,\ldots$ in Remark 2.3.**
>
> **A.** We thank the reviewer for pointing out these and other typographical issues. We have thoroughly proofread the manuscript and correct all such typos (including the ones listed here and those mentioned by other reviewers) to improve readability.

---

> > ### Author Response · Authors · 2025-11-20
> >
> > **Q3: The whole analysis framework is very similar to [1] except for the lemma which interchanges the order of summation (Lemma C.6) to enable the analysis of [1] in the SGDM case. Due to this reason, all results are the same as those of [1]. Is my understanding right? If so, do these results give us any insights? From my perspective, the major innovation of proof is mainly presented in Theorem 3.1 and Theorem 3.3, which is listed in Q3. If any other innovations exist in the proofs of other results compared with [1], please make them clear. If so, I will improve my score.**
> >
> > **A.**
> > We appreciate this opportunity to clarify the relationship with [1].
> > We believe that [1] refers to *High Probability Guarantees for Nonconvex Stochastic Gradient Descent with Heavy Tails* by Li \& Liu (2022).
> > It is true that our analysis framework is similar to [1], but there are many substantive differences once momentum is introduced.
> >
> > *(i) General nonconvex regime (Theorems 3.1 and 3.3).*
> > Compared with the SGD analysis in [1], the SGDM update introduces the coupled momentum variable $\\{m\_t\\}$ so that the stochastic noise no longer appears as a simple martingale sum, but as a *momentum-filtered* process.
> > In particular, our analysis require controlling quantities such as $\sum_{t=1}^{T} \\|m\_t\\|^2$ and $\big\\|\sum_{t=1}^{T}m_t\bigr\\|^2$ through a nontrivial recursion, which has no analogue in the plain-SGD setting of [1].
> >
> > *(ii) PL regime and fast rates (Theorems 3.5, 3.7, 3.9).*
> > The PL-stage analysis for SGDM is considerably more delicate because of the momentum term (as discussed in Appendix D.3).
> > From a technical perspective, the proof of Theorem 3.5 constructs and controls a Lyapunov-type quantity that couples both $x\_t$ and $m\_t$, leading to inequalities of the form
> > $$
> > \begin{aligned}
> > \sum_{t=1}^T \frac{t+t_0-1}{2\mu(S)} \big\\|\nabla F_S(x_t)\big\\|^2
> > & + (T+t_0)(T+t_0-1)\big(F_S(x_{T+1}) - F_S(x_S)\big) \\\\
> > &\le - \sum_{t=1}^T (t+t_0)(t+t_0-1)\gamma
> >    \big\langle m_{t-1}, \nabla F_S(x_{t-1}) \big\rangle   + \sum_{t=1}^{T-1} (t+t_0+1)(t+t_0)L\gamma \\|m_t\\|^2 \\\\
> > &\quad + \sum_{t=1}^T (t+t_0)(t+t_0-1)\frac{L}{2}\\|m_t\\|^2  + (t_0-1)(t_0-2)\big(F_S(x_1) - F_S(x_S)\big) \\\\
> > &\quad - \sum_{t=1}^T (t+t_0)(t+t_0-1)\eta_t
> >    \big\langle \nabla f(x_t; z_{j_t}) - \nabla F_S(x_t), \nabla F_S(x_t) \big\rangle .
> > \end{aligned}
> > $$
> > We then need to carefully bound the cross term
> > \begin{align*}
> >    \sum\_{t=1}^T - (t+t\_0)(t+t\_0-1)\\langle m\_{t-1} , \nabla F\_S(x\_{t-1}) \\rangle,
> > \end{align*}
> > which couples the current momentum to the past gradients, via a recursive argument, together with high-probability bounds on the weighted momentum energy
> > $\sum_{t=1}^T(t+t_0)(t+t_0-1)\\|m_t\\|^2$, in order to obtain a clean $1/T$–type decay.
> > This leads to a key inequality of the form
> > \begin{align*}
> > &\sum_{t=1}^T \frac{t+t_0-1}{2\mu(S)} \big\\|\nabla F_S(x_t)\big\\|^2 - \frac{aK^2}{\mu(S)^2 \alpha} \sum_{t=1}^T (t+t_0-1)^2 \big\\|\nabla F_S(x_t)\big\\|^2 - \frac{L}{2}\sum_{t=1}^T \frac{8C_\gamma}{(1-\gamma)\mu(S)^2} \big\\|\nabla F_S(x_t)\big\\|^2   - L\gamma \sum_{t=1}^{T-1} \frac{8C_\gamma}{(1-\gamma)\mu(S)^2} \big\\|\nabla F_S(x_t)\big\\|^2 \\\\
> > &+ \sum_{t=1}^{T-1} (t+t_0+1)(t+t_0) \eta_t \big\\|\nabla F_S(x_t)\big\\|^2  - \sum_{t=1}^{T-1} L\gamma C_\gamma \frac{8C_\gamma}{(1-\gamma)\mu(S)^2} \big\\|\nabla F_S(x_t)\big\\|^2 - \gamma \frac{a}{\alpha} \sum_{t=1}^{T-1} \bigl(2C_\gamma(t+t_0+1)\mu(S)^{-1}K\bigr)^2 \big\\|\nabla F_S(x_t)\big\\|^2 \\\\
> > &+ (T+t_0)(T+t_0-1)\bigl(F_S(x_{T+1}) - F_S(x(S))\bigr) \\\\
> > &\le
> > L\gamma \frac{8C_\gamma}{(1-\gamma)\mu(S)^2}(T-1)K^2 g(2\theta)\log^{2\theta}(2/\delta)+ \frac{L}{2} \frac{8C_\gamma}{(1-\gamma)\mu(S)^2} T K^2 g(2\theta)\log^{2\theta}(2/\delta) \\\\
> > &+ L\gamma C_\gamma \frac{8C_\gamma}{(1-\gamma)\mu(S)^2}(T-1)K^2 g(2\theta)\log^{2\theta}(2/\delta) + (t_0-1)(t_0-2)\big (F_S(x_1) - F_S(x(S))\big )  + 2\alpha \log(1/\delta) + \gamma 2\alpha \log(1/\delta).
> > \end{align*}
> > These difficulties are specific to SGDM and are not present in the proof techniques of [1].

---

> ### Comment · Reviewer_jh4H · 2025-11-28
>
> Thanks to the authors' detailed replies which address most of my concerns. I am willing to improve my rating to 6. Unfortunately, there is no way to change the rating now. I will appreciate it if PCs, SACs, and ACs consider this comment.

---

### Official Review · Reviewer_tgE3 · 2025-10-31

**Soundness:** 3
**Presentation:** 3
**Contribution:** 3
**Rating:** 6
**Confidence:** 3

**Summary:**

Strengths

1. First high-probability analysis of SGDM in non-convex settings: The paper provides the first comprehensive high-probability convergence and generalization guarantees for stochastic gradient methods with momentum under nonconvex objectives, bridging an important theoretical gap in stochastic optimization.

2. Sharp convergence rates with structural conditions: The results establish an \tilde{O}(1/\sqrt{T}) high-probability convergence rate in the general nonconvex case, improving to \tilde{O}(1/T) under the Polyak–Łojasiewicz condition and even \tilde{O}(1/n^2 + F^*/n) under a Bernstein assumption—showing remarkable theoretical depth.

3. Empirical results align with the theory: Experiments on multiple LIBSVM datasets systematically verify the predicted influence of the tail parameter $\theta$ on convergence speed, offering intuitive visual confirmation of the theoretical high-probability trends.

Weaknesses

1. Limited experimental scope and scalability: Experiments are confined to small-scale logistic regression tasks. The absence of results on large-scale or deep learning benchmarks makes it unclear how well the high-probability bounds manifest in practical training.

2. Lack of practical runtime validation: Theoretical IFO complexity and convergence bounds are not accompanied by empirical runtime or gradient-call comparisons, leaving efficiency gains largely unquantified.

**Strengths:**

see Summary

**Weaknesses:**

see Summary

**Questions:**

NA

---

> ### Author Response · Authors · 2025-11-20
>
> **Dear Reviewer tgE3**, we sincerely appreciate your invaluable and constructive comments, which are very helpful to improve the paper. In light of the reviewers’ comments, we have thoroughly revised the manuscript, carefully going through it almost line by line to improve its clarity, readability, and overall presentation; major revisions are highlighted in blue for ease of reference.
>
> **R1: Limited experimental scope and scalability: Experiments are confined to small-scale logistic regression tasks. The absence of results on large-scale or deep learning benchmarks makes it unclear how well the high-probability bounds manifest in practical training.**
>
> **A**: We thank the reviewer for this insightful comment and fully acknowledge the limited scope of our empirical study. Our submission is intentionally theory-oriented: the main contribution is to provide high-probability convergence and generalization guarantees for SGDM under the general nonconvex settings (and, in particular, under PL and Bernstein-type conditions). The experiment in Appendix A is therefore designed as a sanity check that the tail parameter $\theta$ (sub-Weibull noise) indeed manifests itself in the way predicted by our rates—namely, that larger $\theta$ leads to slower convergence and worse high-probability behavior—rather than as a comprehensive empirical evaluation of SGDM. We fully agree that it would be very valuable to further investigate how well our high-probability bounds manifest in large-scale, practical training problems, such as modern deep-learning benchmarks. We are currently working on a follow-up study that systematically examines this question in more realistic neural-network settings. We believe this will provide a natural companion to the present, theory-focused paper.
>
>
>
> **R2: Lack of practical runtime validation: Theoretical IFO complexity and convergence bounds are not accompanied by empirical runtime or gradient-call comparisons, leaving efficiency gains largely unquantified.**
>
> **A**: We thank the reviewer for raising this point. Our work does not propose a new optimization algorithm; instead, it provides sharper high-probability convergence and generalization guarantees for the *standard* SGDM. The IFO complexity bounds appear as a theoretical corollary of our rates, translating them into an oracle-complexity form, and are not meant to claim improved practical efficiency over existing SGDM implementations. For this reason, we did not include additional empirical runtime or gradient-call comparisons.

---

### Official Review · Reviewer_ctUE · 2025-10-31

**Soundness:** 2
**Presentation:** 1
**Contribution:** 2
**Rating:** 4
**Confidence:** 3

**Summary:**

The paper studies high-probability convergence and generalization bounds for SGDM in non-convex settings, achieving faster rates under the Polyak-Łojasiewicz (PL) and a Bernstein-type gradient condition.
Specifically, assuming that the stochastic noise follows a sub-Weibull distribution parameterized by $\theta$, the authors establish a slightly tighter bound at $\theta = 1/2$ (improving the logarithmic factor from $\log(T/\delta)$ to $\log(1/\delta)$) and extend the analysis to the case $\theta > 1$, compared with prior work.

**Strengths:**

1. The paper provides convergence and generalization bounds for SGDM under arbitrary values of $\theta$, offering a unified treatment that covers a wide range of noise distributions from sub-Gaussian to heavy-tailed cases.

2. The focus on SGDM rather than plain SGD is interesting, as understanding how momentum affects convergence and generalization remains an important open question in stochastic optimization.

**Weaknesses:**

1. For the case $\theta = 1/2$, the improvement over related work is essentially from $\log(T/\delta)$ to $\log(1/\delta)$ while keeping the same leading-order rate. This represents a mild tightening rather than a substantive advance.

2. The extension to the case $\theta > 1$ relies on general concentration inequalities applicable to arbitrary $\theta$ (Appendix Lemmas C.2–C.4). It seems that the main change lies in using a more general tail inequality, which does not introduce substantial technical difficulty or genuine novelty.

3. The paper is difficult to follow, with many typos and inconsistencies. Examples include:

   Line 161: “smothness” → “smoothness”.
   Assumption 2.1: The definition of smoothness should not be embedded in the assumption itself.
   Theorems: All numbering “(1.) (2.) (3.) (4.)” should be corrected to “(1). (2). (3). (4).”.

   Line 286: “study” should be “studied”.
   Theorem 5: The term $\mu(S)$ appears before being defined.
   Line 373: “theFS” should be “the FS”.
   Table 1: The first two rows list identical assumptions but yield different error bounds—this should be clarified.


To enhance coherence and credibility, I recommend that the authors thoroughly revise the paper for both clarity and presentation quality.

**Questions:**

1. Under the same assumptions as in the main theorems, are there corresponding high-probability convergence and generalization results for SGD (without momentum)? A direct comparison would help clarify whether momentum offers any provable benefit in this setting.

2. Which concrete model classes (e.g., deep neural networks) are known to satisfy the PL condition?

---

> ### Author Response · Authors · 2025-11-20
>
> **Dear Reviewer ctUE**, we sincerely appreciate your invaluable and constructive comments, which are very helpful to improve the paper. In light of the reviewers’ comments, we have thoroughly revised the manuscript, carefully going through it almost line by line to improve its clarity, readability, and overall presentation; major revisions are highlighted in blue for ease of reference.
>
> **R1: For the case $\theta = 1/2$, the improvement over related work is essentially from $\log (T/\delta)$ to $\log(1/\delta)$ while keeping the same leading-order rate. This represents a mild tightening rather than a substantive advance.**
>
> **A**:
> We thank the reviewer for this thoughtful comment. Although this improvement is only logarithmic, it may be the strongest
> possible refinement in the general nonconvex setting we consider. For smooth nonconvex stochastic optimization with a first-order oracle and controlled noise, the rate $\mathcal{O}(1/\sqrt{T})$ in terms of the expected squared gradient norm is known to be optimal (up to logarithmic factors) [1]. Consequently, under the same structural
> assumptions, any further progress can only affect constants and logarithms but
> not the leading $1/\sqrt{T}$ scaling.
>
> [1] Yossi Arjevani, Yair Carmon, John C Duchi, Dylan J Foster, Nathan Srebro, and Blake Woodworth.
> Lower bounds for non-convex stochastic optimization. arXiv preprint arXiv:1912.02365, 2019.
>
>
> **R2: The extension to the case $\theta >1$ relies on general concentration inequalities applicable to arbitrary $\theta$ (Appendix Lemmas C.2–C.4). It seems that the main change lies in using a more general tail inequality, which does not introduce substantial technical difficulty or genuine novelty.**
>
> **A**: We thank the reviewer for this comment. The general concentration inequalities
> (Appendix Lemmas C.2–C.4) are indeed a fundamental tool, but the extension to
> $\theta>1$ is not a simple plug-in of a more general tail bound.
>
> - *For Theorem 3.1.*
>   Extending the analysis from the sub-Gaussian case
>   $\theta = \tfrac{1}{2}$ in Li \& Orabona (2020) to general sub-Weibull
>   $\theta \ge \tfrac{1}{2}$ is not mechanical. In the Freedman-type inequality
>   (Lemma C.4), when $\theta > \tfrac{1}{2}$ the parameter $\alpha$ must satisfy
>   a nontrivial constraint $\alpha \ge b \max_t m_t$. As a result, we must
>   carefully *jointly* calibrate $\alpha$ and the stepsizes
>   $\\{\eta_t\\}_{t=1}^T$ so that the stochastic terms remain dominated by the
>   deterministic descent while still achieving the optimal
>   $\mathcal{O}(1/\sqrt{T})$ scaling.
> - *Beyond Theorem 3.1: fast rates and generalization.*
>   Compared with Li \& Orabona (2020), our PL-based convergence and
>   generalization bounds are genuinely new. Li \& Orabona do not obtain fast
>   rates, whereas our fast-rate analysis (technically involved as discussed in Appendix D.3) goes
>   beyond their framework. Moreover, Theorems 3.3, 3.7, and 3.9 provide,
>   to the best of our knowledge, the first *high-probability*
>   generalization guarantees for SGDM in the nonconvex setting. A key ingredient
>   is a new way of controlling the iterate norm $\\|\mathbf{x}_t\\|$ (and thus the
>   effective complexity of the process) under different structural assumptions;
>   this technique does not appear in Li \& Orabona (2020) and may be of
>   independent interest for future analyses.

---

> > ### Author Response · Authors · 2025-11-20
> >
> > **R3: The paper is difficult to follow, with many typos and inconsistencies.
> > Examples include:
> > Line 161: “smothness” → “smoothness”.
> > Assumption 2.1: The definition of smoothness should not be embedded in the assumption itself.
> > Theorems: All numbering “(1.) (2.) (3.) (4.)” should be corrected to “(1). (2). (3). (4).”.
> > Line 286: “study” should be “studied”. Theorem 5: The term $\mu(S)$
> >  appears before being defined. Line 373: “the$F_S$” should be “the $F_S$”.
> > Table 1: The first two rows list identical assumptions but yield different error bounds—this should be clarified. To enhance coherence and credibility, I recommend that the authors thoroughly revise the paper for both clarity and presentation quality.**
> >
> > **A**: We thank the reviewer for carefully pointing out these concrete issues with clarity and presentation. We have thoroughly revised the manuscript to improve readability and fix all reported problems, including (but not limited to):
> >
> > - correcting the typos ("smothness'' $\rightarrow$ "smoothness'',  "study'' $\rightarrow$  "studied'',  "the$F_S$'' $\rightarrow$  "the $F_S$'', etc.);
> > - restructuring the smoothness assumption;
> > - correcting the theorem numbering format  "(1).(2).(3).(4).'' to the standard  "(1). (2). (3). (4).'';
> > - defining $\mu(S)$ in Remark 2.8 before it is first used in Theorem 5;
> > - clarifying Table 1: In Appendix B, we provided some explainations of Table 1 and clarified this difference. The two rows under reference [1] correspond to two *different* results from Li \& Orabona (2020): the first is for the standard SGDM analyzed there, while the second is for a delayed AdaGrad-with-momentum variant whose stepsize does not depend on the current gradient. We have also rewrote Appendix B for both clarity and presentation quality.
> >
> > **Q1: Under the same assumptions as in the main theorems, are there corresponding high-probability convergence and generalization results for SGD (without momentum)? A direct comparison would help clarify whether momentum offers any provable benefit in this setting.**
> >
> > **A**: A work [1] derives high-probability convergence and generalization results for *SGD without momentum* under the same assumptions, yielding rates of the same order as those obtained here (up to constants and mild logarithmic factors). This paper *closes the theoretical gap for SGDM*: we show that the widely used momentum method, under the general nonconvex / PL / Bernstein assumptions, also enjoys high-probability convergence and generalization guarantees of comparable order to those known for SGD, so that SGDM has essentially the same theoretical performance as SGD under these conditions.
> >
> > [1] Shaojie Li and Yong Liu. High probability guarantees for nonconvex stochastic gradient descent
> > with heavy tails. In International Conference on Machine Learning, pp. 12931–12963, 2022.
> >
> > **Q2: Which concrete model classes (e.g., deep neural networks) are known to satisfy the PL condition?**
> >
> > **A**: Thanks for this review. Many important models are known to satisfy a PL inequality, at least locally. Notable examples satisfying the PL
> > condition include
> > two-layer neural networks [1], matrix completion [2], dictionary learning [3], and phase retrieval [4]. [5] provide empirical evidence that the (smoothed) loss of practical deep networks locally exhibits a one-point convexity property of PL type. More rigorously, [6] analyze
> > over-parameterized shallow networks with quadratic activations and prove that,
> > in the interpolation regime where the training loss is zero, the empirical
> > risk satisfies a PL inequality. These examples motivate our focus on studying SGDM under the PL  curvature assumption.
> >
> > [1] Yuanzhi Li and Yang Yuan. Convergence analysis of two-layer neural networks with relu activation.
> > In Advances in Neural Information Processing Systems, pages 597–607, 2017.
> >
> > [2] Ruoyu Sun and Zhi-Quan Luo. Guaranteed matrix completion via non-convex factorization. IEEE
> > Transactions on Information Theory, 62(11):6535–6579, 2016.
> >
> > [3] Sanjeev Arora, Rong Ge, Tengyu Ma, and Ankur Moitra. Simple, efficient, and neural algorithms for
> > sparse coding. Journal of Machine Learning Research (JMLR), 2015.
> >
> > [4] Yuxin Chen and Emmanuel Candes. Solving random quadratic systems of equations is nearly as easy
> > as solving linear systems. In Advances in Neural Information Processing Systems, pages 739–747,
> > 2015.
> >
> > [5] Robert Kleinberg, Yuanzhi Li, and Yang Yuan. An alternative view: When does sgd escape local
> > minima? arXiv preprint arXiv:1802.06175, 2018.
> >
> > [6] Mahdi Soltanolkotabi, Adel Javanmard, and Jason D Lee. Theoretical insights into the optimization
> > landscape of over-parameterized shallow neural networks. IEEE Transactions on Information Theory,
> > 2018.

---

### Official Review · Reviewer_WfMC · 2025-11-04

**Soundness:** 4
**Presentation:** 3
**Contribution:** 3
**Rating:** 6
**Confidence:** 4

**Summary:**

The authors have obtained high probability gradient norm bounds in a general non-convex problem for stochastic gradient descent with momentum (SGDM). Then under PL condition of loss, they show high probability bounds on the function values and achieve faster rates under Bernstein condition for gradients.

**Strengths:**

The authors have done a very good job in terms of literature review. The paper is overall very well written. The assumptions are clearly provided and the theoretical results are well stated. Establishing both convergence and generalization bounds in high probability with clear description is very interesting.

**Weaknesses:**

This is a nice submission. While I note that the authors have focused on theory, providing a toy experiment in Appendix A will be the main weakness of this submission. I think the paper will be significantly improved by providing more relevant experiments with clear connection with the developed rates under various assumptions and potentially in a more practical setting.

**Questions:**

Please check the weaknesses

---

> ### Author Response · Authors · 2025-11-20
>
> **Dear Reviewer WfMC**, we sincerely appreciate your invaluable and constructive comments, which are very helpful to improve the paper. In light of the reviewers’ comments, we have thoroughly revised the manuscript, carefully going through it almost line by line to improve its clarity, readability, and overall presentation; major revisions are highlighted in blue for ease of reference.
>
> **R: This is a nice submission. While I note that the authors have focused on theory, providing a toy experiment in Appendix A will be the main weakness of this submission. I think the paper will be significantly improved by providing more relevant experiments with clear connection with the developed rates under various assumptions and potentially in a more practical setting.**
>
> **A**:
> We thank the reviewer for this insightful comment. Our submission is intentionally theory-oriented: the main contribution is to provide high-probability convergence and generalization guarantees for SGDM under the general nonconvex settings (and, in particular, under PL and Bernstein-type conditions). The experiment in Appendix A is therefore designed as a sanity check that the tail parameter $\theta$ (sub-Weibull noise) indeed manifests itself in the way predicted by our rates—namely, that larger $\theta$ leads to slower convergence and worse high-probability behavior—rather than as a comprehensive empirical evaluation of SGDM. We fully agree that more extensive experiments on large-scale, practical tasks—designed explicitly to probe the connection between the developed rates and the various assumptions—would be very valuable. We are currently working on a follow-up study that is dedicated to a more extensive empirical investigation of the theory developed here in practical deep-learning settings. We believe this will provide a natural companion to the present, theory-focused paper.

---

### Author Response · Authors · 2025-12-01

Dear PCs, SACs, ACs, and Reviewers,

We would first like to sincerely thank you for the time and effort you have devoted to handling and reviewing our submission, as well as for the many constructive suggestions that helped us improve the paper. In particular, we are deeply grateful to the ACs, SACs, and PCs for their heavy workload and careful coordination in what we understand has been an especially challenging year. We hope that this brief global response can help you quickly grasp our main clarifications and thereby ease some of the load on your side.

Based on the first-round reviews, the two main concerns about our work were: (1) the limited scope of the experiments (Reviewers WfMC and tgE3), and (2) the novelty of the proof techniques underlying our theoretical results (R1 and R2 of Reviewer ctUE and Q3 of Reviewer jh4H).

Regarding experiments, our submission is intentionally theory-oriented: the core contribution is to establish high-probability convergence and generalization guarantees for SGDM under broad nonconvex settings. The toy experiments in Appendix A were designed only as sanity checks that our sub-Weibull tail parameter behaves in practice as predicted by the theory, and were not intended to be a central contribution of the main text. Motivated by the reviewers’ feedback, we are currently working on a follow-up, experiment-driven study devoted to a systematic empirical study of SGDM.

Regarding novelty, we responded theorem by theorem to clarify the new ideas in our analysis. We detailed how our arguments go beyond prior techniques once momentum and sub-Weibull noise are introduced, and how the proof structure and key inequalities differ from existing analyses. Reviewer jh4H explicitly acknowledged this novelty. We believe that our results provide a meaningful and, in some aspects, essentially best-possible improvement over Li & Orabona (2020) within this framework, and that both the fast rates and, especially, the high-probability generalization bounds fill an important gap in the current literature on SGDM.

Finally, we have thoroughly revised the manuscript, going through it almost line by line to improve clarity, readability, and overall presentation. Major revisions and structural changes are highlighted in blue for ease of reference.

We once again thank the PCs, SACs, ACs, and Reviewers for their efforts and understanding this year, and we would be very happy to continue the conversation if further clarification is helpful.

---

### Meta-Review · Area_Chair_HVkK · 2025-12-26

**Summary:**

In this paper, the authors have obtained high-probability gradient norm bounds in a general non-convex problem for SGD with momentum (SGDM). Then under the typical PL condition of the loss function, they also show high probability bounds on the function values and achieve faster rates under Bernstein condition for the gradient noise. in particular, it seems that this is the first high-probability generalization guarantees for SGDM in the nonconvex setting.

The scores for this submission is 6644. However, it seems to me that Reviewer jh4H in the message gave score 4 are very satisfied with the response from the author and mentioned that they are willing to increase the score to 6. In this case, the final scores are 6664 which clearly indicates that the majority of the reviewers are positive about this submission.  Hence, I recommend its acceptance.

**Reviewer Concerns:**

the concerns are mostly addressed in the rebuttal by the authors. in particular, Reviewer jh4H who originally gave score 4 is very satisfied with the response from the author and mentioned that they are willing to increase the score to 6.

**Reviewer Scores:**

Yes, Reviewer jh4H in the message gave score 4 are very satisfied with the response from the author and mentioned that they are willing to increase the score to 6.

---

### Decision · Program_Chairs · 2026-01-26

Accept (Poster)